# Progressive divisions of multipotent neural progenitors generate late-born chandelier cells in the neocortex

Khadeejah T. Sultan[1,2], Wenying Angela Liu[1,3], Zhao-Lu Li[4], Zhongfu Shen[4], Zhizhong Li[1], Xin-Jun Zhang[1], Owen Dean[1], Jian Ma[4] & Song-Hai Shi[1,2,3]

Diverse γ-aminobutyric acid (GABA)-ergic interneurons provide different modes of inhibition to support circuit operation in the neocortex. However, the cellular and molecular mechanisms underlying the systematic generation of assorted neocortical interneurons remain largely unclear. Here we show that NKX2.1-expressing radial glial progenitors (RGPs) in the mouse embryonic ventral telencephalon divide progressively to generate distinct groups of interneurons, which occupy the neocortex in a time-dependent, early inside-out and late outside-in, manner. Notably, the late-born chandelier cells, one of the morphologically and physiologically highly distinguishable GABAergic interneurons, arise reliably from continuously dividing RGPs that produce non-chandelier cells initially. Selective removal of Partition defective 3, an evolutionarily conserved cell polarity protein, impairs RGP asymmetric cell division, resulting in premature depletion of RGPs towards the late embryonic stages and a consequent loss of chandelier cells. These results suggest that consecutive asymmetric divisions of multipotent RGPs generate diverse neocortical interneurons in a progressive manner.

[1] Developmental Biology Program, Sloan Kettering Institute, Memorial Sloan Kettering Cancer Center, 1275 York Avenue, New York, NY 10065, USA. [2] Neuroscience Graduate Program, Weill Cornell Medical College, 1300 York Avenue, New York, NY 10065, USA. [3] BCMB Graduate Program, Weill Cornell Medical College, 1300 York Avenue, New York, NY 10065, USA. [4] School of Life Sciences, Tsinghua-Peking Joint Center for Life Sciences, IDG/McGovern Institute for Brain Research, Tsinghua University, 100084 Beijing, China. These authors contributed equally: Khadeejah T. Sultan, Wenying A. Liu. Correspondence and requests for materials should be addressed to J.M. (email: majian0509@mail.tsinghua.edu.cn) or to S.-H.S. (email: shis@mskcc.org)

The neocortex consists of glutamatergic excitatory neurons and GABAergic inhibitory interneurons. While glutamatergic neurons generate the main output of neural circuits, diverse populations of GABAergic interneurons provide a rich array of inhibition that regulates circuit operation[1,2]. Neocortical interneurons are incredibly diverse in their morphology, molecular marker expression, membrane and electrical properties, and synaptic connectivity[3,4]. While the rich variety of interneuron subtypes endows the inhibitory system with the requisite power to shape circuit output across a broad dynamic range, little is known about the cellular and molecular mechanisms underlying the systematic generation of diverse neocortical interneuron populations.

Most of our understanding of neocortical neurogenesis has come from studies of excitatory neuron production. Derived from neuroepithelial cells, radial glial cells in the developing dorsal telencephalon account for the major neural progenitor cells that generate virtually all neocortical excitatory neurons[5–7]. They reside in the ventricular zone (VZ) with a characteristic bipolar morphology and actively divide at the luminal surface of the VZ. At the early stage (i.e., before embryonic day 11-12, E11-12, in mice), radial glial progenitors (RGPs) largely undergo symmetric proliferative division to amplify the progenitor pool. After that, RGPs predominantly undergo asymmetric neurogenic division to self-renew and simultaneously produce neurons either directly or indirectly via transit amplifying progenitor cells such as intermediate progenitors (IPs) or outer subventricular zone RGPs (oRGs, also called basal RGPs or intermediate RGPs) that further divide in the subventricular zone (SVZ). The orderly division behavior of RGPs essentially determines the number and types of excitatory neurons constituting the neocortex.

Previous studies have provided important insights into the mechanisms that allow for the generation of a rich array of neuronal types from a given progenitor population. One mechanism involves a common pool of progenitors that continuously undergoes asymmetric neurogenesis and becomes progressively fate-restricted over time, thereby generating distinct neuronal subtypes at different times. This is the case for the principal neuronal types found in the vertebrate retina[8–10]. The other mechanism is via multiple pools of fate-restricted progenitors that may be spatially, temporally, or molecularly segregated so as to produce distinct neuronal types, such as the developing spinal cord, where different populations of neurons arise from progenitors expressing distinct transcription factors[11].

In the case of excitatory neurons in the neocortex, several lines of evidence suggest that diversity is established predominantly via the first mechanism described above; that is, excitatory neurons in different layers of the neocortex with distinct properties and functions are sequentially generated from a common pool (i.e., multipotent) of RGPs that undergoes progressive fate restriction[12–16]. Notably, a recent study suggested that a subpopulation of RGPs exclusively generates superficial layer excitatory neurons, raising the possibility of fate-restricted RGPs in neocortical excitatory neurogenesis[17]. However, subsequent studies argued against the proposed fate-restricted RGP model[18–21]. Nonetheless, these studies point to the importance of understanding progenitor behavior in the context of the generation of diverse neuronal types. This is especially pertinent for neocortical interneurons, as the developmental mechanisms and logic of their production at the progenitor level are not well understood.

Over 70% of neocortical inhibitory interneurons are derived from the homeodomain transcription factor NKX2.1-expressing progenitor cells located in the transient regions of the ventral telencephalon known as the medial ganglionic eminence (MGE) and the preoptic area (PoA)[22–28]. Among the diverse collection of neocortical interneurons, chandelier (or axo-axonic) cells are considered to be a bone fide subtype[29–33]. They selectively target the axon initial segment (AIS) of postsynaptic cells with characteristic candlestick-like arrays of axonal cartridges, and thus control pyramdial cell activity through the release of GABA. Recent genetic and transplantation studies showed that neocortical chandelier cells are selectively generated by NKX2.1-expressing progenitor cells in the MGE/PoA at the late embryonic stage[34,35]. However, it remains unclear whether chandelier cells originate from a common pool of multipotent neural progenitors or a specified (i.e., fate-restricted) pool of neural progenitors in the MGE/PoA.

In this study, we selectively labeled dividing RGPs in the MGE/PoA at different embryonic stages and systematically examined their interneuron output in the neocortex. As development proceeds, dividing RGPs produce distinct groups of interneuron progeny that exhibit an initial inside-out and late outside-in pattern in laminar distribution. Interestingly, chandelier cells are generated at a reliable rate at the late embryonic stage by continuously dividing RGPs that produce non-chandelier cells at the early embryonic stage. Disruption of RGP asymmetric division by selective removal of the evolutionarily conserved cell polarity protein Partition defective 3 (PARD3) causes premature depletion of RGPs and a consequent loss of chandelier cells in the neocortex. These results support a progressive fate specification program via consecutive asymmetric divisions of multipotent MGE/PoA RGPs in generating diverse neocortical interneurons, among which chandelier cells are the last output.

## Results

**Selective labeling of dividing MGE/PoA RGPs and their output**. To examine the progenitor origin of neocortical interneuron diversity (Supplementary Fig. 1), we first assessed interneuron output by dividing RGPs in the MGE/PoA at different embryonic time points. We took advantage of a previously established method that combines mouse genetics and in utero retroviral injection to selectively label dividing RGPs at the VZ surface of the MGE/PoA[36] (Supplementary Fig. 2a). This method harnesses the exquisite fidelity of the interaction between the subgroup A avian sarcoma leukosis virus (ASLV) and its cognate receptor TVA which is not expressed in mammalian tissues.

By setting up timed pregnancies between the $R26^{LSL-TVAiLacZ}$ knock-in mouse line that expresses TVA in a Cre recombinase-dependent manner[37] and the $Nkx2.1$-$Cre$ transgenic mouse line[22], in which the Cre recombinase is selectively expressed in the vast majority of RGPs of the MGE/PoA except those in the dorsal most edge (Supplementary Fig. 2b), we generated embryos that specifically expressed the TVA receptor in NKX2.1+ MGE/PoA RGPs, rendering them susceptible to avian retrovirus infection. Dividing NKX2.1+ RGPs at the VZ surface of the MGE/PoA were then labeled in a temporally specific manner by performing in utero intraventricular injections of high-titer RCAS (replication competent ASLV long terminal repeat with splice acceptor) retroviruses expressing enhanced green fluorescent protein (EGFP) at a defined embryonic stage (i.e., E12, E13, E14, E15, E16, or E17) (Supplementary Fig. 2a)[36]. We focused our analysis at E12-E17, when the vast majority of MGE/PoA-derived neocortical interneurons are generated. The same amount of RCAS-EGFP retrovirus was injected at each embryonic stage, and numerous RGPs were selectively infected and thereby labeled in the VZ of the MGE/PoA, but not the lateral ganglionic eminence (LGE) or the neocortex (NCX) (Supplementary Fig. 2c). The brains of the injected mice were analyzed at P21, when interneuron neurogenesis, migration, apoptosis, and

differentiation are largely complete. Notably, this strategy is more accurate for examining the temporal neocortical interneuron output of dividing MGE/PoA RGPs than the typical inducible genetic fate mapping approach using *Nkx2.1-CreER;Ai9-tdTomato* mice, in which labeled interneurons arise from both dividing and non-dividing (i.e., divide at a late time point) RGPs, as well as IPs in the SVZ at the time of tamoxifen (TM) induction as NKX2.1 is expressed in MGE/PoA RGPs and IPs regardless of cell cycle phase at the embryonic stage[22,23,38,39]. In comparison, RCAS retrovirus only infects and labels dividing RGPs at the VZ surface in the MGE/PoA.

EGFP-expressing interneurons were abundantly found in the neocortex, hippocampus, and striatum, possessing characteristic morphological features (Supplementary Fig. 2d) and molecular marker expression such as parvalbumin (PV) and somatostatin (SOM) (Supplementary Fig. 3), as previously shown[22,23,36]. EGFP-expressing glial cells with distinct morphology were also found in the subcortical structures but not in the neocortex at P21 (Supplementary Fig. 2d, broken rectangle and Supplementary Fig. 2e). To systematically examine the number and distribution of labeled interneurons, we performed serial sectioning, immunohistochemistry, and three-dimensional (3D) reconstruction of the three main forebrain structures populated by the Nkx2.1+ MGE/PoA-derived interneurons, including the neocortex, hippocampus, and striatum (Supplementary Fig. 2f). Based on the 3D stereological analysis[36,40], we estimated the total number and distribution of interneurons produced by dividing NKX2.1+ MGE/PoA RGPs labeled at different embryonic stages, with a particular focus on the neocortex, which harbors the vast majority (~80–90%) of their total interneuron output[40].

**Progressive changes in neocortical interneuron output**. We systematically assessed the neocortical interneuron output of dividing NKX2.1+ MGE/PoA RGPs at different embryonic stages from E12 to E17 (Fig. 1). Notably, the labeling efficiency at each embryonic stage across different animals was highly consistent (Supplementary Fig. 4). Interestingly, both the number and distribution of interneuron progeny in the neocortex exhibited dynamic changes (Fig. 1a–c). We quantitatively analyzed the interneuron output in the different neocortical layers, which can be recognized based on cellular organization and the Allen Brain Atlas, across the entire neocortex. As the labeling time shifted from E12 to E17, the total interneuron output progressively decreased (Fig. 1a–c). The vast majority of neocortical interneurons were produced by dividing NKX2.1+ MGE/PoA RGPs at E12-E15. On the other hand, a relatively small but reliable number of neocortical interneurons were produced by dividing NKX2.1+ MGE/PoA RGPs at E16-E17.

Interestingly, while the interneuron progeny of dividing NKX2.1+ MGE/PoA RGPs labeled at E12 were broadly distributed in both the deep and superficial layers, they became gradually restricted as the time of labeling proceeded to the later stages (Fig. 1a, b, d). Between E12-E15, the interneuron output progressively shifted from the deep to superficial layers, consistent with a time-dependent inside-out pattern of generation. In sharp contrast, between E15-E17, the interneuron output of dividing RGPs gradually shifted from the superficial to deep layers, indicative of a time-dependent outside-in pattern of production. Together, these results suggest that NKX2.1+ MGE/PoA RGPs produce neocortical interneurons in a temporally bimodal fashion of laminar occupation, that is initially inside-out and then outside-in. While the inside-out interneuron genesis from the MGE/PoA has been suggested previously[41–46], the late outside-in pattern of interneuron genesis has not been described before.

We next examined the representative subtypes of neocortical interneuron progeny in the primary somatosensory area generated by dividing NKX2.1+ MGE/PoA RGPs labeled at different embryonic stages on the basis of the morphological features. At E12, the vast majority of labeled interneurons were Martinotti cells with characteristic long ascending layer 1 projecting axons, basket cells, dense arbor cells with local, dense neurite arbors, and descending axon cells with descending layer 5/6 projecting axons in the deep and superficial layers (Supplementary Fig. 5a, left and 5b). In addition, we observed a small number of chandelier cells with characteristic axonal cartridges in the superficial layer 2 and deep layers 5 and 6, and horizontal cells with dense horizontally oriented neurite arbors along the white matter boundary in layer 6 (Supplementary Fig. 5a, left, red line squares and 5b). At E14, the majority of labeled interneurons were Martinotti cells, basket cells, dense arbor cells, and descending axon cells in superficial layers, in addition to a small number of chandelier cells in the superficial layer 2, and a small number of chandelier cells and horizontal cells in the deep layers (Supplementary Fig. 5a, middle and 5b). At E16, the majority of labeled interneurons were chandelier cells in the superficial layer 2, and chandelier cells and horizontal cells in the deep layers (Supplementary Fig. 5a, right and 5b). PV-expressing and SOM-expressing interneurons account for the two major subtypes of neocortical interneurons originated from the MGE/PoA. We found that, as development proceeded, while the relative generation of PV+ interneurons by dividing NKX2.1+ RGPs progressively increased, the relative generation of SOM+ interneuron gradually decreased, especially after E14 (Supplementary Fig. 6). In addition, we found that Martinotti cells and dense arbor cells are largely SOM+, whereas basket cells and chandelier cells are mostly PV+ (Supplementary Fig. 7), consistent with previous studies[2,47–51]. Taken together, these results suggest that different subtypes of neocortical interneurons are reliably and progressively generated by dividing NKX2.1+ MGE/PoA RGPs at different embryonic stages. Moreover, these results indicate that Martinotti cells, basket cells, dense arbor cells and decending axon cells are generated in an inside-out manner, whereas chandelier cells, as well as horizontal cells are generated in an outside-in fashion (Supplementary Fig. 5c).

**Dividing RGPs at E12 generate distinct interneurons**. The temporal changes in the number, subtype, and laminar distribution of interneurons output by dividing Nkx2.1+ MGE/PoA RGPs at different embryonic stages may reflect the progressive change of a common multipotent RGP pool or distinct fate-restricted populations of RGPs that preferentially divide at different embryonic time points (Supplementary Fig. 1). To distinguish between these two possibilities, we examined the birth date of the interneuron progeny generated by dividing Nkx2.1+ MGE/PoA RGPs at E12 (Fig. 2). EdU and BrdU were administered at E12 and E15, respectively, to the pregnant females that received in utero intraventricular RCAS-EGFP injection at E12 (Fig. 2a). EdU-labeled and BrdU-labeled EGFP-expressing interneurons in the neocortex were analyzed at P21.

EGFP-expressing interneurons were found across the deep and superficial layers of the neocortex (Fig. 2b, left). EdU-labeled cells were predominantly in the deep layers, whereas BrdU-labeled cells were mostly in the superficial layers (Fig. 2b, right). EdU and BrdU administered at E12 and E15 labeled distinct pools of EGFP-expressing interneuron populations in the neocortex (Fig. 2b). While the majority of EGFP-expressing interneurons in layer 6 were labeled by EdU, the majority of EGFP-expressing interneurons in layer 2/3 were instead labeled by BrdU (Fig. 2c). In addition, the majority of EGFP-expressing interneurons

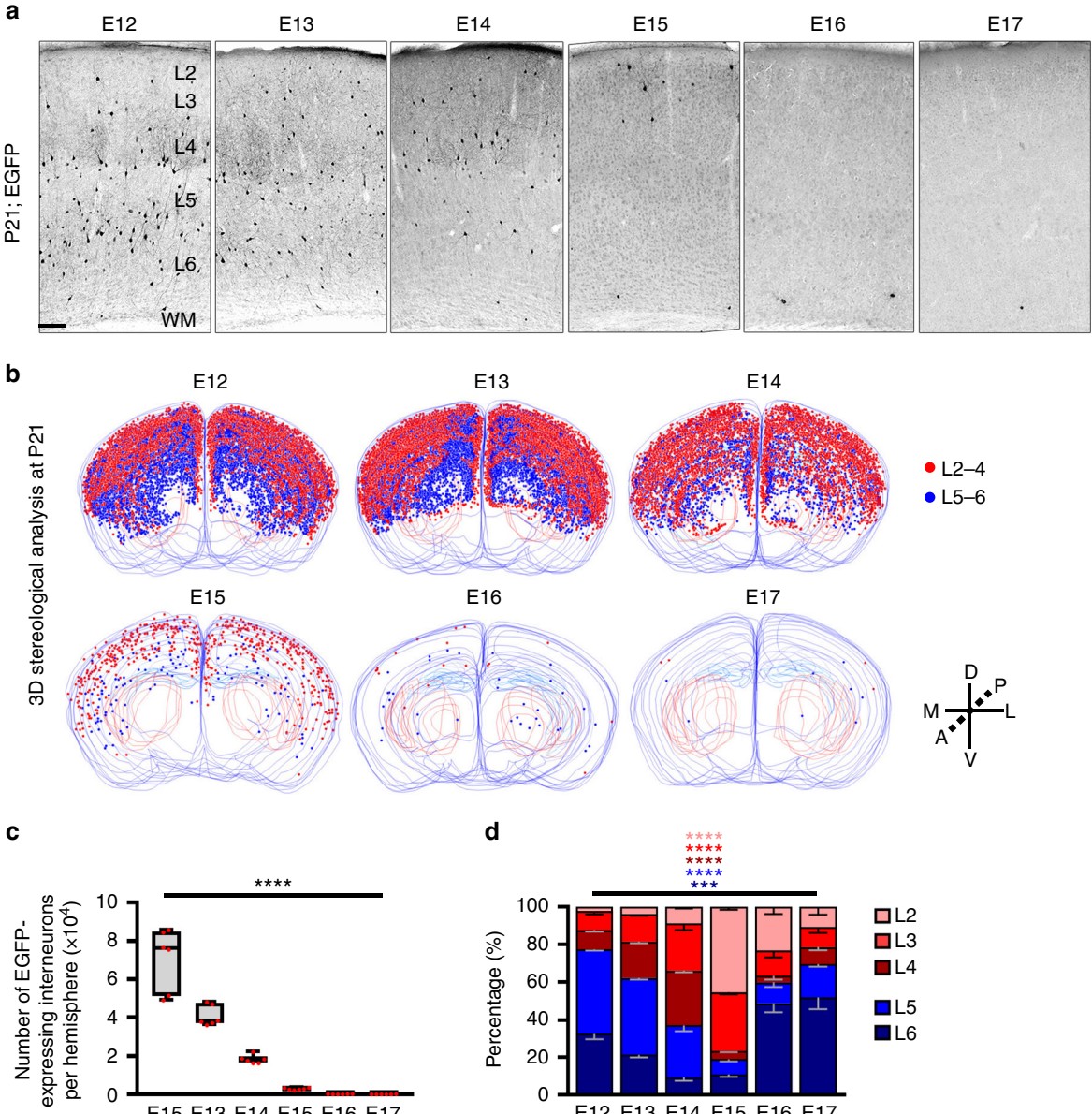

**Fig. 1** Progressive changes in neocortical interneuron output of dividing NKX2.1+ MGE/PoA RGPs. **a** Images of the somatosensory cortex of representative P21 mouse brains that received in utero intraventricular injection of high-titer EGFP-expressing retrovirus at E12-E17. EGFP-labeled interneurons are shown in black. L, layer; WM, white matter. Scale bar: 150 μm. **b** 3D reconstruction images of P21 brains that received in utero retrovirus injection at different embryonic stages. Blue lines indicate the contours of the whole brain. Red and blue dots represent the cell bodies of the EGFP-expressing interneurons in the superficial (L2-4) and deep (L5/6) layers of the neocortex, respectively. A anterior, P posterior, D dorsal; V ventral. **c** Quantification of the total number of EGFP-expressing interneurons per hemisphere of the neocortex. ($n = 6$ hemispheres per time point from at least three different animals). Center line, median; box, interquartile range; whiskers, minimum and maximum. Red dots indicate individual data points. ****$P < 0.0001$ ($P = 3.4e-13$, Jonckheere–Terpstra test). **d** Quantification of the laminar distribution of EGFP-expressing interneurons. Data are presented as mean ± SEM ($n = 6$ hemispheres per time point from at least three different animals). *** $P = 0.0001$; **** $P < 0.0001$ (Kruskal-Wallis test)

in layer 4 were not labeled by EdU or BrdU. Together, these results suggest that dividing Nkx2.1+ MGE/PoA RGPs at E12 produce distinct neocortical interneurons at different time points, with deep layer interneurons generated early and superficial layer interneurons generated late, consistent with the previous genetic fate mapping analysis in the ventral eminences using the *Olig2-CreER* transgenic mice[41] (Supplementary Fig. 8a). Our data suggest that dividing Nkx2.1+ MGE/PoA RPGs at E12 undergo multiple rounds of divisions at different embryonic stages to produce interneurons occupying different layers of the neocortex.

**Dividing RGPs at E12 generate chandelier cell late**. Having found that dividing Nkx2.1+ MGE/PoA RGPs at E12 divide progressively and generate distinct neocortical interneurons at different time points, we next explored the identities of these interneurons. Interestingly, we reliably observed EGFP-expressing chandelier cells with the characteristic morphology at the border between layers 1 and 2, where a major population of chandelier cells are located[35], in the P21 neocortices that received in utero intraventricular retrovirus injection at E12 (Fig. 3a). They possessed numerous characteristic vertical cartridges in layers 2/3 (Fig. 3a, insets). Whereas only 3.6 ± 0.6% of the total EGFP-

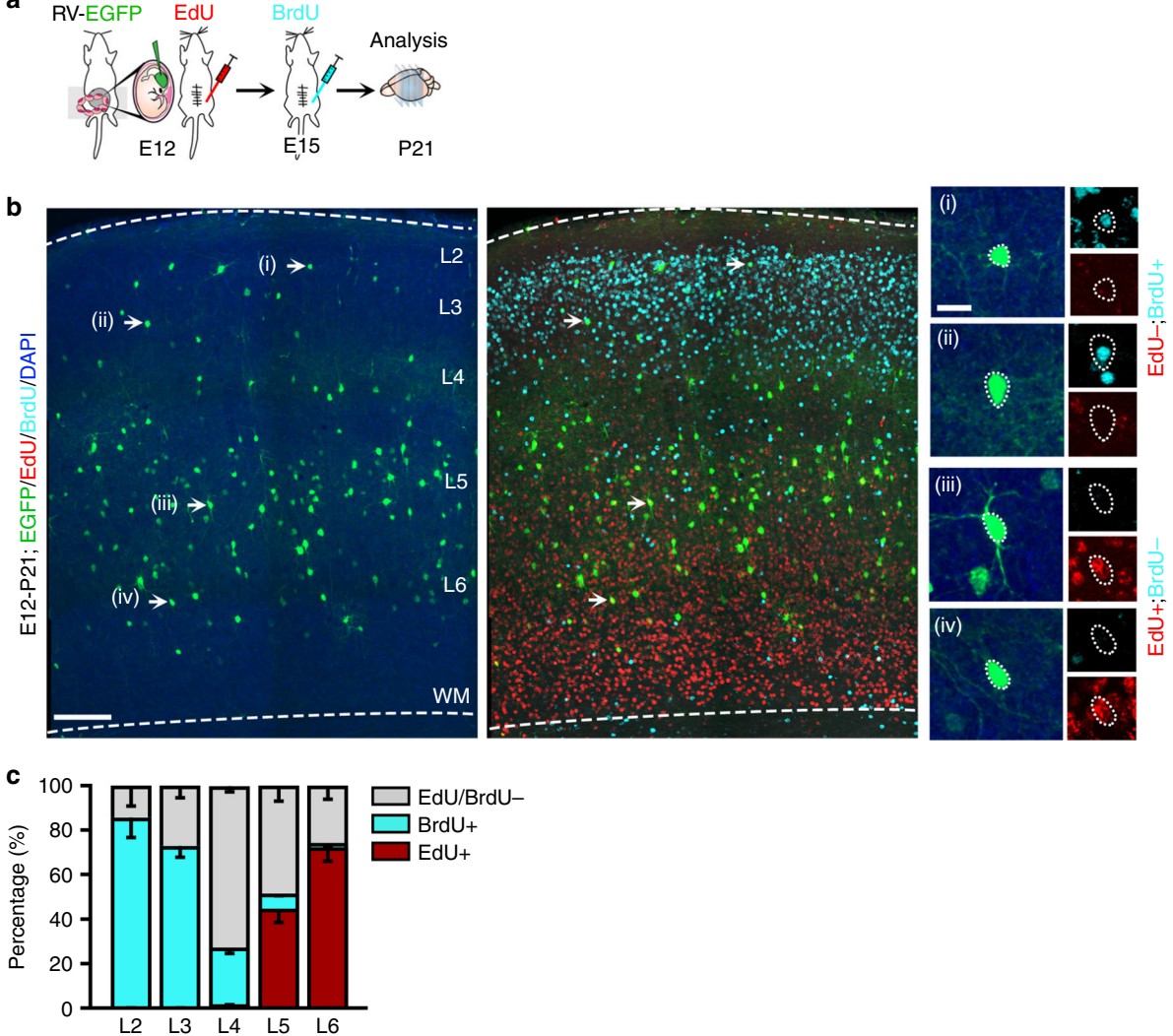

**Fig. 2** Dividing NKX2.1$^+$ MGE/PoA RGPs at E12 generate distinct neocortical interneurons at different time points. **a** Schematic of the experimental design. Animals received in utero injection of EGFP-expressing retrovirus, as well as EdU injection at E12 and BrdU injection at E15 were analyzed at P21. **b** Image of the somatosensory cortex of P21 mouse brain stained for EGFP (green), EdU (red), and BrdU (green), and counterstained with DAPI (blue). Broken lines represent the pial surface and white matter boundaries. L layer, WM white matter. High magnification images of representative EGFP-expressing interneurons in different layers (arrows) are shown to the right. Scale bars: 150 μm and 25 μm. **c** Percentage of EGFP-expressing interneurons in different layers labeled with EdU or BrdU. Data are presented as mean ± SEM ($n = 3$ brains)

expressing interneuron output was identified as chandelier cells (Fig. 3b, c), nearly half (48.1 ± 1.2%) of EGFP-expressing interneurons within layer 2 were chandelier cells (Fig. 3d, e). These results clearly suggest that dividing Nkx2.1$^+$ RGPs labeled at E12 generate superficial layer chandelier cells in the neocortex.

We next examined the birth date of labeled chandelier cells originating from dividing Nkx2.1$^+$ MGE/PoA RGPs at E12. To achieve this, we administered two consecutive pulses of EdU 4 h apart to pregnant dams at E13, E14, or E15 that received in utero intraventricular retrovirus injection at E12 (Fig. 3f). We then analyzed the percentage of EGFP-expressing superficial layer chandelier cells that were labeled by EdU at different embryonic stages (Fig. 3g, h), which reflected their birth date. While the percentage of EGFP-expressing superficial layer chandelier cells labeled by EdU was very low at E13 (0.0 ± 0.0%) and E14 (16.7 ± 8.3%), it reached 93.3 ± 6.7% at E15 (Fig. 3h). These results suggest that the vast majority of superficial layer chandelier cells are produced at the late embryonic stage, even though they originate from dividing Nkx2.1$^+$ MGE/PoA RGPs labeled at E12.

In comparison, the EGFP-expressing interneurons labeled by EdU at E13 and E14 were predominantly non-chandelier cells located in the deep layers (Supplementary Fig. 9). Together, these results suggest that dividing Nkx2.1$^+$ RGPs at E12 initially generate non-chandelier cells in the deep layers and later generate chandelier cells, as well as some non-chandelier cells in the superficial layers.

**Consecutive divisions of RGPs generate chandelier cell**. The progressive generation of distinct interneurons at different embryonic stages by dividing Nkx2.1$^+$ MGE/PoA RGPs at E12 suggests that RGPs undergo consecutive divisions to produce different interneurons, with chandelier cells being the late-born ones. To directly test this, we performed sequential retrovirus labeling experiments by injecting mCherry-expressing retrovirus at E12 and EGFP-expressing retrovirus at E14, and analyzed the brains at P21 (Fig. 4a and Supplementary Fig. 10). As shown above, mCherry-expressing retrovirus injected at E12 labeled

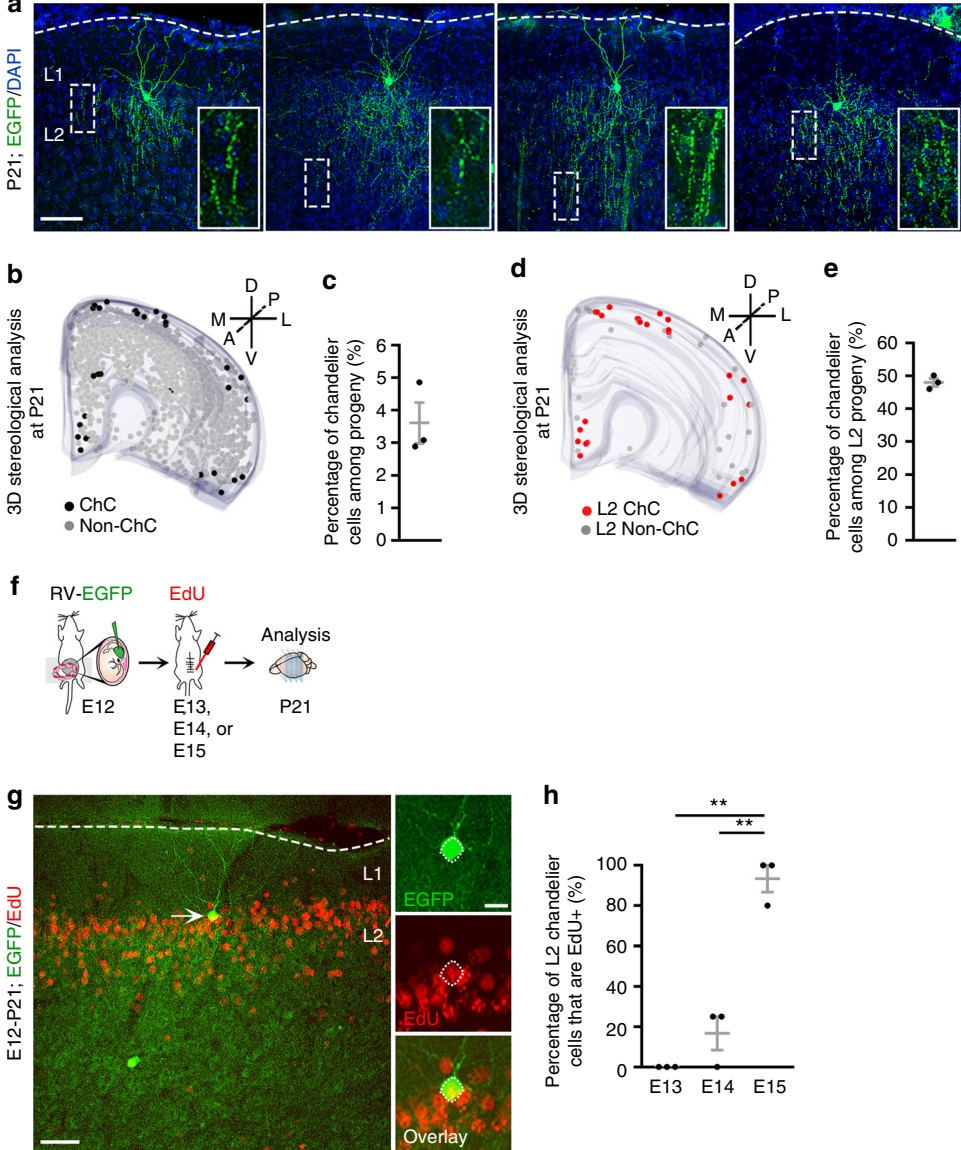

**Fig. 3** Dividing NKX2.1[+] MGE/PoA RGPs at E12 generate superficial layer chandelier cells at the late embryonic stage. **a** Images of P21 neocortices injected with EGFP-expressing retrovirus at E12, stained for EGFP (green), and counterstained with DAPI (blue). EGFP-expressing chandelier cells were frequently found at the border of superficial layers 1 and 2. High magnification images of the vertical arrays of axonal cartridges characteristic of the chandelier cell (broken rectangles) are shown in the insets. Broken lines indicate the pial surface. Scale bar: 50 μm. **b** 3D reconstruction image of a neocortical hemisphere showing EGFP-expressing chandelier cells and non-chandelier cells. Black and gray dots represent the cell bodies of EGFP-expressing chandelier and non-chandelier cells, respectively. D dorsal, P posterior, L lateral. **c** Percentage of chandelier cells among EGFP-expressing neocortical interneurons. Gray lines represent mean ± SEM. Black dots represent individual brains ($n = 3$). **d** 3D reconstruction image of a neocortical hemisphere showing EGFP-expressing chandelier and non-chandelier cells in layer 2 only. Red and gray dots represent the cell bodies of EGFP-expressing chandelier and non-chandelier cells, respectively. **e** Percentage of chandelier cells among EGFP-expressing interneurons in L2. Gray lines indicate mean ± SEM. Black dots represent individual brains ($n = 3$). **f** Schematic of the experimental design for birth dating analysis of EGFP-expressing chandelier cells. Animals received in utero retrovirus injection at E12 and EdU at E13, or E14, or E15 were analyzed at P21. **g** Image of P21 neocortex injected with EGFP-expressing retrovirus at E12 followed by EdU injection at E15, stained for EGFP (green) and EdU (red). The arrow indicates an EGFP-expressing chandelier cell labeled by EdU injection at E15. High magnification images of the chandelier cell are shown to the right. The broken line indicates the pial surface. Scale bars: 150 μm and 10 μm. **h** Percentage of EGFP-expressing chandelier cells that are labeled by EdU. Gray lines represent mean ± SEM. Black dots represent individual brains ($n = 3$ per time point). **P = 0.005 (E13-E15); **P = 0.002 (E14-E15) (unpaired t-test with Welch's correction)

interneurons spanning both the deep and superficial layers (Fig. 4b, red), whereas EGFP-expressing retrovirus injected at E14 predominantly labeled interneurons mostly in the superficial layers, and some scattered ones in the deep layers (Fig. 4b, green). Interestingly, the chandelier cells at the layer 1-2 border expressed both EGFP and mCherry (Fig. 4b, insets and Supplementary

Fig. 10), indicating that the RGPs from which the chandelier cells originated are infected by both mCherry-retroviruses and EGFP-retroviruses injected at E12 and E14, respectively. These results demonstrate that the same RGPs undergo consecutive divisions at E12 and E14 in the process of generating late-born superficial layer chandelier cells.

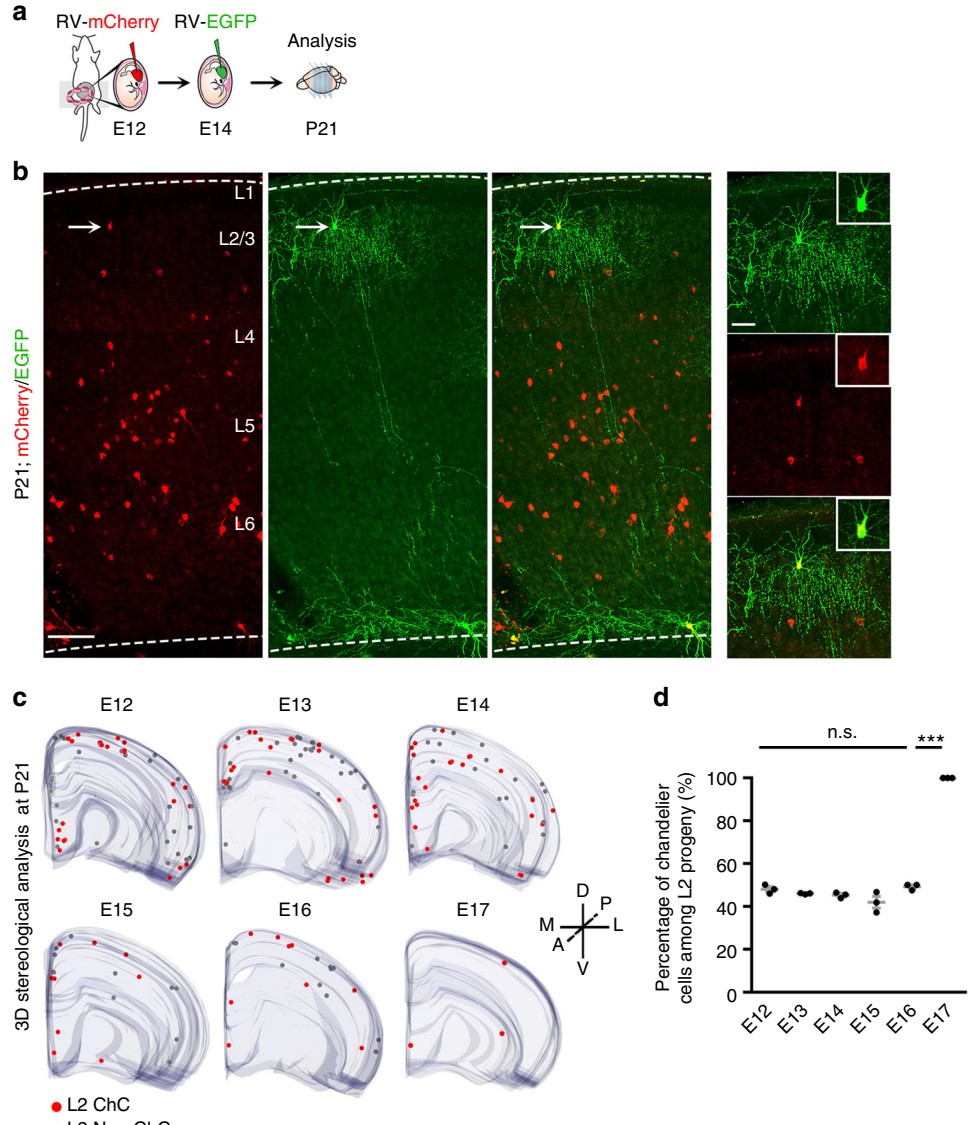

**Fig. 4** Consecutive divisions of NKX2.1$^{+}$ MGE/PoA RGPs generate superficial layer chandelier cells. **a** Schematic of the experimental design. Animals received in utero injection of mCherry-retroviruses and EGFP-retroviruses at E12 and E14, respectively, were analyzed at P21. **b** Images of P21 neocortex stained for mCherry (red) and EGFP (green). The arrows indicate a superficial layer chandelier cell expressing both mCherry and EGFP. High magnification images of the chandelier cells are shown to the right. Broken lines indicate the pial surface and white matter (WM) boundaries. L, layer. Scale bars: 150 μm and 50 μm. **c** 3D reconstruction images of the neocortical hemispheres that received in utero retrovirus injection at E12-E17. Red and gray dots represent EGFP-expressing chandelier and non-chandelier cells in L2, respectively. A anterior, P posterior, D dorsal, V ventral, M medial, L lateral. **d** Percentage of chandelier cells among EGFP-expressing interneurons in L2. Gray lines represent mean ± SEM. Black dots represent individual brains (n = 3 per time point). ***P = 0.0003 (E16 vs. E17); n.s., not significant (E12 vs. E13, P = 0.2258; E13 vs. E14, P = 0.4920; E14 vs. E15, P = 0.3375; E15 vs. E16, P = 0.1069); unpaired t-test with Welch's correction)

The consecutive RGP division and the preferential generation of superficial layer chandelier cells at the late embryonic stage raise the intriguing possibility that chandelier cell production is a temporally programmed property of Nkx2.1$^{+}$ MGE/PoA RGPs. In other words, chandelier cell production is a predictable and defined property of Nkx2.1$^{+}$ MGE/PoA RGPs as long as they reach certain (i.e., late) embryonic stage regardless of the initial labeling time by retrovirus. Moreover, the overall rate of chandelier cell production at the late embryonic stage (i.e., the percentage of chandelier cells among all late-born superficial layer interneuron progeny) by dividing Nkx2.1$^{+}$ MGE/PoA RGPs labeled at different early embryonic stages should be relatively constant. To test this, we systematically examined the fraction of

superficial layer interneurons generated by dividing Nkx2.1$^{+}$ MGE/PoA RGPs labeled at different embryonic stages that were chandelier cells (Fig. 4c, d). Remarkably, we found that dividing Nkx2.1$^{+}$ MGE/PoA RGPs labeled between E12 and E16 generated a similar or constant proportion (~50%) of chandelier cells among all interneuron progeny in the layer 2 (Fig. 4d), even though the absolute number of chandelier cell progeny progressively decreased, especially after E14 (Supplementary Fig. 11a, b), as expected from the decrease in the total interneuron output (Fig. 1). At E17, virtually all labeled cells in the superficial layer 2 were chandelier cells, indicating that chandelier cells are the last interneuron output by Nkx2.1$^{+}$ MGE/PoA RGPs. These results suggest that the capacity for chandelier cell production at

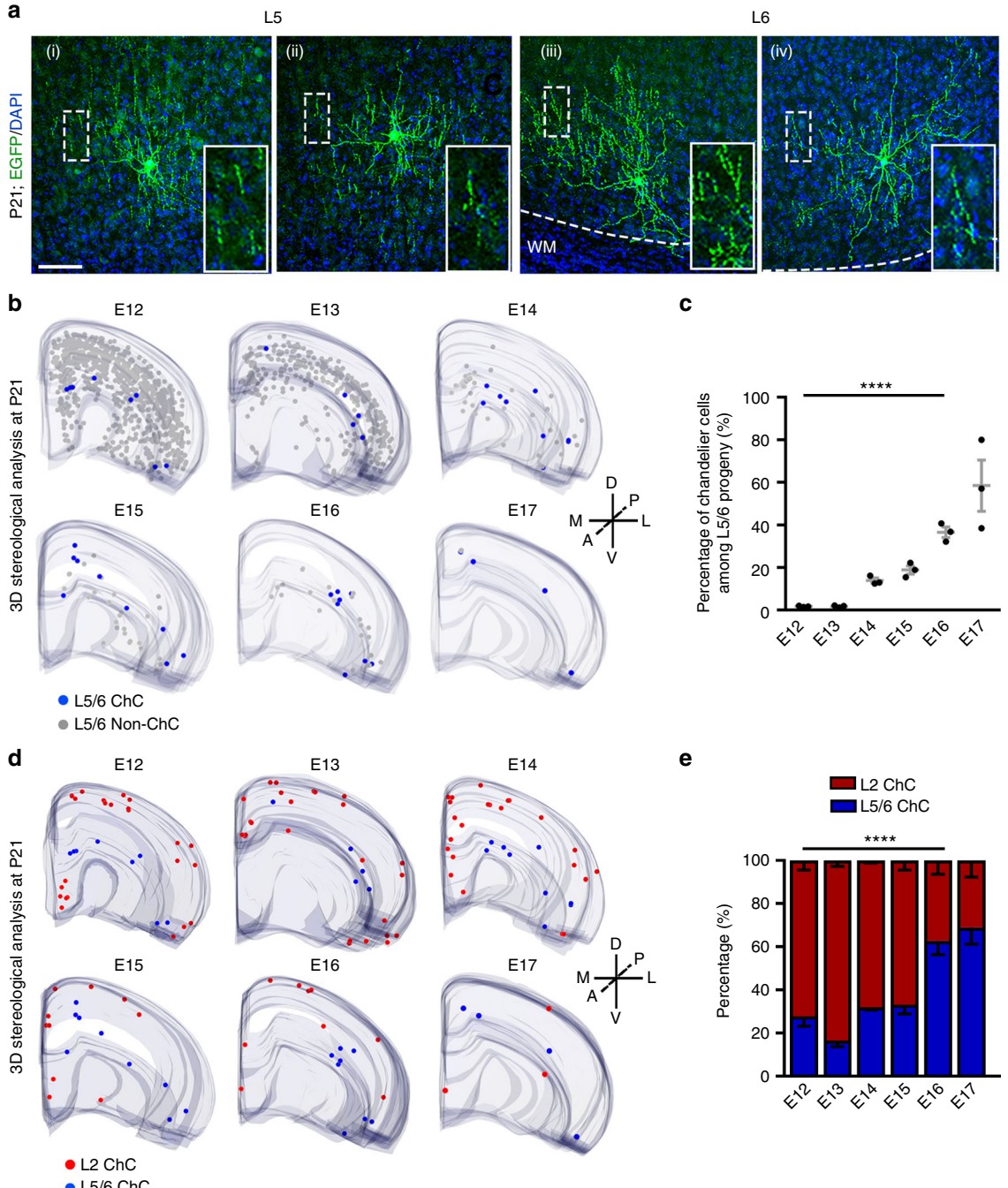

**Fig. 5** Late outside-in neurogenesis by NKX2.1+ MGE/PoA RGPs generates deep layer chandelier cells. **a** Images of P21 neocortices that received *in utero* EGFP-expressing retrovirus injection, stained for EGFP (green), and counterstained with DAPI (blue). EGFP-expressing chandelier cells were found in layers 5 (left two cells) and 6 (right two cells). High magnification images of the vertical arrays of axonal cartridges characteristic of the chandelier cell (broken rectangles) are shown in the insets. Broken lines represent the white matter (WM) boundary. Scale bar: 50 μm. **b** 3D reconstruction images of the neocortical hemispheres showing EGFP-expressing chandelier cells and non-chandelier cells in layers 5/6. Blue and gray dots represent the cell bodies of L5/6 chandelier and non-chandelier cells, respectively. **c** Percentage of chandelier cells among EGFP-expressing interneurons in layers 5/6. Gray lines represent mean ± SEM. Black dots represent individual brains ($n = 3$ per time point). ****$P < 0.0001$ (one-way ANOVA). **d** 3D reconstruction images of the neocortical hemispheres showing EGFP-expressing chandelier cells in layers 2 and 5/6. Red and blue dots represent the cell bodies of EGFP-expressing chandelier cells in the superficial layer 2 and deep layers 5/6, respectively. A anterior, P posterior, D dorsal; V ventral, M medial, L lateral. **e** Ratios of EGFP-expressing chandelier cells in the superficial layer 2 and deep layers 5/6. ****$P < 0.0001$ (one-way ANOVA)

the late embryonic stage is an inherent property of Nkx2.1+ MGE/PoA RGPs. Regardless when Nkx2.1+ MGE/PoA RGPs are labeled (e.g., E12 vs. E14), as development proceeds, their cumulative potential of generating layer 2 chandelier cells is defined and predicable.

**Late interneuron genesis produces deep layer chandelier cell.** In addition to superficial layer chandelier cells, we also observed EGFP-expressing chandelier cells in layers 5 and 6 of the neocortex with characteristic cartridge structures (Fig. 5a). The frequency of labeling deep layer chandelier cells by in utero

retrovirus injection at early embryonic stages was very low (E12: 1.7 ± 0.2%; E13: 1.7 ± 0.3%), as the vast majority of labeled interneurons were non-chandelier cells (Fig. 5b, c). As the labeling time proceeded towards the late embryonic stage, the fraction of labeled deep layer interneurons identified as chandelier cells progressively increased (E16: 36.6 ± 2.5%; E17: 58.5 ± 12.0%) (Fig. 5b, c). Notably, this increase was largely due to a decrease in the generation of non-chandelier cells in the deep layers, whereas the number of labeled chandelier cells in the deep layers remained largely comparable between E12-E16 (Fig. 5b). The relatively constant output of deep layer chandelier cells across different embryonic stages indicates that deep layer chandelier cell production is also a programmed feature of Nkx2.1$^+$ MGE/PoA RGPs. Sequential retrovirus labeling showed that deep layer chandelier cells, as well as layer 6 horizontal cells were also generated by consecutively dividing RGPs at the late embryonic stage (Supplementary Fig. 11c–e). Together, these results suggest that Nkx2.1$^+$ MGE/PoA RGPs undergo consecutive divisions and progressively generate different neocortical interneurons at different embryonic stages. At the early embryonic stage, they produce predominantly non-chandelier cells in the deep layers. As time proceeds, while the output of non-chandelier cells is progressively reduced, the generation of superficial and deep layer chandelier cells occurs at a predictable and constant rate.

Given that both superficial and deep layer chandelier cells were generated at the late embryonic stage, we examined the relative number of superficial versus deep layer chandelier cells produced by dividing Nkx2.1$^+$ MGE/PoA RGPs labeled at different embryonic stages (Fig. 5d, e). Interestingly, the ratio of superficial versus deep layer chandelier cells was comparable between E12-E15. During this peak phase of interneuron neurogenesis, there were more superficial layer than deep layer chandelier cells labeled. In comparison, at E16 and E17, there were more deep layer than superficial layer chandelier cells labeled. These results suggest that deep layer chandelier cells are largely generated after superficial layer chandelier cells. Notably, this shift in superficial versus deep layer chandelier cell generation coincided with the time-dependent outside-in laminar distribution of late-generated neocortical interneurons (Fig. 1d and Supplementary Fig. 5b). Collectively, these results suggest that, while relatively small in number, late interneuron neurogenesis after E15 produces specific interneuron subtypes such as deep layer chandelier cells, as well as layer 6 horizontal cells to regulate neocortical circuit function.

**PARD3 regulates NKX2.1$^+$ RGP asymmetric division**. Consecutive RGP divisions in the process of generating neocortical chandelier cells indicate that MGE/PoA RGPs undergo multiround asymmetric divisions, consistent with our previous time-lapse imaging analysis of individual MGE/PoA RGPs in organotypic brain slice culture[36]. PARD3, an evolutionarily conserved cell polarity protein, has been shown to play an essential role in controlling RGP asymmetric division in the developing neocortex[52,53]. To test whether PARD3 regulates asymmetric division of MGE/PoA RGPs, we generated a conditional *Pard3* mutant mouse line, *Pard3$^{fl/fl}$*, and crossed it to the *Nkx2.1-Cre* mouse line[22] (Supplementary Fig. 12a). While PARD3 was abundantly expressed in RGPs, especially at the VZ surface, in the wild type MGE, it was depleted in the VZ of the *Nkx2.1-Cre; Pard3$^{fl/fl}$* conditional knockout (referred to as *Pard3* cKO hereafter) MGE at E11 (Supplementary Fig. 12b).

To examine the effect of PARD3 removal on MGE/PoA RGPs, we stained the wild type and *Pard3* cKO brains with an antibody against OLIG2, a transcription factor highly expressed in RGPs of the ventral telencephalon including the MGE/PoA[54] (Supplementary Fig. 8b, c). We found that the number of OLIG2$^+$ RGPs in the MGE/PoA was progressively reduced in the *Pard3* cKO brain compared with the wild type brain at the late embryonic stages (Fig. 6a, b), suggesting that removal of PARD3 leads to a progressive and drastic loss of RGPs in the MGE/PoA towards the late embryonic stages.

Asymmetric division of RGPs ensures an intricate balance between RGP renewal/maintenance and neurogenesis. The loss of MGE/PoA RGPs in the *Pard3* cKO brain may be due to a defect in RGP asymmetric division. To directly test this, we performed in vivo clonal analysis to explicitly assess the division mode of individually dividing RGPs in the MGE/PoA (Fig. 6c). We injected serially diluted, low-titer retroviruses expressing EGFP into the lateral ventricle at E14 and recovered the brain at E15 for analysis. To examine the division mode of sparsely labeled RGPs in the MGE/PoA, brains were serially sectioned and stained with antibodies against OLIG2 and Ki67, a proliferative cell marker. We identified all sparsely labeled cell pairs in the MGE/PoA that originated from individual dividing RGPs by 3D reconstruction (Fig. 6d, e and Supplementary Movies 1 and 2). In these experiments, OLIG2$^+$/Ki67$^+$, OLIG2$^-$/Ki67$^+$, and OLIG2$^-$/Ki67$^-$ cells corresponded to RGPs, IPs, and post-mitotic interneurons (INs), respectively. As expected, the majority (~56.5%, 26 out of 46) of the cell pairs in the wild type control MGE contained a bipolar RGP and a multi-polar IP or IN (Fig. 6d, e left and 6f), indicating asymmetric division. In contrast, only a small fraction (~19.5%, 8 out of 41) of the cell pairs in the *Pard3* cKO MGE contained a bipolar RGP and a multi-polar IP or IN; instead, the vast majority (~68.3%, 28 out of 41) of the cell pairs contained two IPs or two INs (Fig. 6d, e right and 6f), indicating symmetric terminal division. Together, these results suggest that PARD3 removal leads to a switch in RGP division mode from asymmetric neurogenic division (one RGP and one IP or IN) to symmetric terminal division (i.e., two IPs or two INs), which would result in a loss of MGE/PoA RGPs.

**Chandelier cell production depends on asymmetric division**. We next examined the consequence of impaired RGP asymmetric division on chandelier cell production (Fig. 7). We performed in utero intraventricular injection of Cre-dependent EGFP-expressing retrovirus into the wild type control and *Pard3* cKO brains at E15 (Fig. 7a). Brains were collected at P21 and subjected to 3D stereological analysis to systematically examine EGFP-expressing chandelier cells in the neocortex (Fig. 7b–g). While the control brain contained 16.0 ± 4.0 chandelier cells per neocortical hemisphere (14.8 ± 3.0% of EGFP-expressing population), the *Pard3* cKO brain contained only ~6.0 ± 1.0 chandelier cells per neocortical hemisphere (5.5 ± 0.8% of EGFP-expressing population) (Fig. 7b, c).

We further examined the superficial and deep layer chandelier cell output. There was a significant reduction in the number of labeled chandelier cells in layer 2 per neocortical hemisphere in the *Pard3* cKO brain (4.0 ± 3.0; 19.2 ± 2.4% of EGFP-expressing population in layer 2) compared with the control brain (10.0 ± 3.0; 37.8 ± 7.3% of EGFP-expressing population in layer 2) (Fig. 7d, e). Similarly, the number of labeled chandelier cells in layers 5/6 significantly decreased from 6.0 ± 2.0 (14.7 ± 4.2% of EGFP-expressing population in layers 5/6) in the control brain to 2.0 ± 0.5 (4.1 ± 1.3% of EGFP-expressing population in L5/6) in the *Pard3* cKO brain (Fig. 7f, g). Together, these results clearly suggest that MGE/PoA RGP asymmetric division is crucial for the generation of both superficial and deep layer chandelier cells.

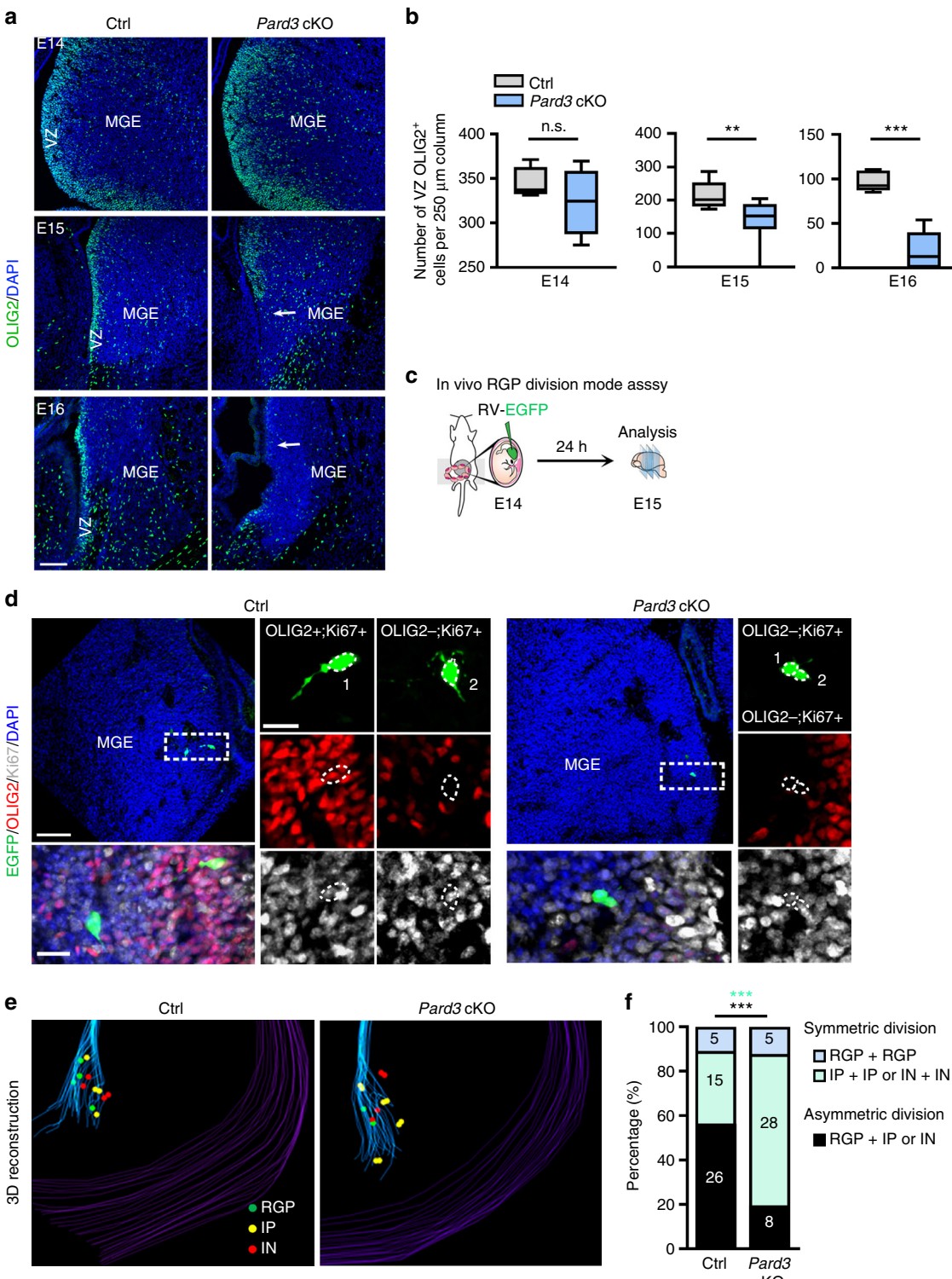

## Discussion

Extensive efforts have been made to delineate different types of interneurons in the neocortex; yet, little is known about the developmental program of generating diverse neocortical interneurons. Over 70% of neocortical interneurons, including non-overlapping PV-expressing (e.g., basket and chandelier) and SOM-expressing (e.g., Martinotti) cells, are generated by NKX2.1[+] RGPs in the MGE/PoA[22,24,26,55]. The remaining 30% of neocortical interneurons such as vasoactive intestinal peptide (VIP)-expressing cells are produced by RGPs in the CGE[55–57]. Recent single-cell RNA-seq analyses suggested that there are on the order of ~20 transcriptomically distinct interneurons in the neocortex[58,59]. While it is well-established that RGPs in the MGE/PoA and CGE give rise to distinct neocortical interneurons, it remains largely unclear how RGPs in the MGE/PoA or CGE produce different subtypes of neocortical interneurons. By systematically and selectively examining the neocortical interneuron output of NKX2.1[+] MGE/PoA RGPs dividing at different

**Fig. 6** PARD3 regulates NKX2.1[+] MGP/PoA RGP asymmetric division. **a** Images of E14, E15, and E16 MGE/PoA in control (Ctrl) and *Pard3* cKO mouse brains stained for OLIG2 (green) and counterstained with DAPI (blue). The arrows indicate the loss of OLIG2[+] cells in the ventricular zone (VZ) of *Pard3* cKO MGE. Scale bar: 100 μm. **b** Quantification of the number of OLIG2[+] cells in the VZ per 250 μm column in the MGE/PoA (E14: $n = 4$ sections from 3 control brains and $n = 4$ sections from 3 *Pard3* cKO brains; E15: $n = 7$ sections from 3 control brains and $n = 10$ sections from 7 *Pard3* cKO brains; E16: $n = 6$ sections from 3 control brains and $n = 6$ sections from 3 *Pard3* cKO brains). Center line, median; box, interquartile range; whiskers, minimum and maximum. \*\*$P < 0.01$, \*\*\* $P < 0.001$, n.s., not significant($P = 0.38$) (unpaired t-test with Welch's correction). **c** Schematic of in vivo clonal analysis to assess RGP division pattern. **d** Images of E15 control and *Pard3* cKO mouse brains injected with low-titer EGFP-expressing retrovirus (green) at E14 and stained for Ki67 (white) and OLIG2 (red), and counterstained with DAPI (blue). High magnification images of the EGFP-expressing cell pairs (broken rectangles) are shown to the bottom and right. Scale bars: 50 μm, 20 μm, and 15 μm. **e** 3D reconstruction images of control and *Pard3* cKO MGE/PoA showing EGFP-expressing cell pairs containing RGP (green), IP (yellow), or interneurons (IN, red). **f** Percentage of EGFP-expressing cell pairs representing symmetric proliferative division, asymmetric neurogenic division, or symmetric terminal division in the MGE/PoA of control ($n = 6$) and *Pard3* cKO ($n = 6$) brains. \*\*\* $P = 0.0004$ (black); \*\*\* $P = 0.0009$ (cyan) (Chi-square test)

embryonic stages, we found that MGE/PoA RGPs divide progressively to produce distinct neocortical interneurons. Dividing MGE/PoA RGPs at E12 produce neocortical interneurons that occupy both the superficial and deep layers, with the deep layer interneurons mostly born early and the superficial layer interneurons born late. As development proceeds, dividing MGE/PoA RGPs produce neocortical interneurons with more restricted laminar identity. Our data points to a multipotent pool of MGE/PoA RGPs that undergoes consecutive asymmetric cell divisions to progressively produce different subtypes of neocortical interneurons, with chandelier cells among the last output.

While the laminar identity and morphological features are key properties of neocortical interneurons, they may not directly correspond to defined interneuron subtypes. To assess the relationship between RGP division behavior and interneuron subtype specification, we focused on chandelier cells, a bona-fide interneuron subtype[29–33]. The highly characteristic arrays of axonal boutons into individual cartridges allow explicit identification of chandelier cells based on the morphology. Interestingly, dividing MGE/PoA RGPs at E12 reliably give rise to chandelier cells located at the border of layers 1 and 2, in addition to a large number of non-chandelier cells in the deep layers. Notably, these chandelier cells are not born at E12, but at the late embryonic stage (e.g., E15), suggesting that dividing MGE/PoA RGPs labeled at E12 undergo additional rounds of divisions after the initial division to produce superficial layer chandelier cells at the later time point. We confirmed this by performing sequential retrovirus injection experiment, wherein deep layer non-chandelier cells were labeled by retrovirus injected at E12, whereas superficial layer chandelier cells were double-labeled by retroviruses sequentially injected at E12 and E14. These results explicitly demonstrate that MGE/PoA RGPs undergo consecutive divisions at E12 and E14 in the process of producing superficial layer chandelier cells. Furthermore, these results suggest that, even though chandelier cells are highly distinctive with a well-defined developmental origin spatially and temporally[34,35], they are not originated from a fate-restricted or pre-specified pool of RGPs in the MGE/PoA.

Remarkably, the proportion of chandelier cells among superficial layer interneuron progeny generated by dividing MGE/PoA RGPs labeled at E12-E16 was relatively constant. These results suggest that, regardless of the labeling time and density, the production of superficial layer chandelier cells is a programmed and predicable feature of MGE/PoA RGPs. In other words, as development proceeds, dividing MGE/PoA RGPs undergo several rounds of divisions to produce non-chandelier cells before producing superficial layer chandelier cells at the later time point. It is likely that distinct interneurons are generated at different rounds of RGP division in conjunction with intermediate progenitor production. Future efforts to systematically delineate this program, as well as the underlying molecular regulation will

provide fundamental new insights into neocortical interneuron generation and diversity.

Supplementing the previously suggested inside-out generation of neocortical interneurons[41–46], our data revealed a previously unrecognized outside-in generation of neocortical interneurons by MGE/PoA RGPs at the late embryonic stage. While the vast majority of MGE/PoA-derived neocortical interneurons are generated at E12-E15, a relatively small but significant number of neocortical interneurons are generated at E16-E17, including chandelier cells in the superficial and deep layers, and horizontal cells in layer 6. Interestingly, the late-born interneurons exhibit a time-dependent outside-in distribution in the neocortex. The vast majority of deep layer neocortical interneurons are generated at the early embryonic stage; yet, some deep layer neocortical interneurons, in particular chandelier cells, are generated at the late embryonic stage. The additional wave of deep layer interneuron production suggests that these late-born deep layer interneurons are likely distinct from those generated earlier and thereby play unique roles in modulating deep layer neuron activity and function. Chandelier cell terminals are particularly prominent in layers 5 and 6 of the human temporal neocortex[60,61] and impaired function of chandelier cells is thought to be a key component in the etiology of human epilepsy[62]. Our data suggest that dividing MGE/PoA RGPs at E12-17 progress through a temporal program in generating diverse neocortical interneurons at different embryonic stages. NKX2.1 expression in the MGE/PoA begins by ~E10 with onset of neurogenesis. Dividing MGE/PoA RGPs prior to E12 likely exhibit a similar temporal interneuron neurogenic program in producing non-chandelier cells initially and chandelier cells later. It is possible that RGPs starting the temporal neurogenic program earlier (e.g., undergo neurogenic division at E10 or E11 to produce interneurons) also complete the program (i.e., exit the cell cycle or neurogenic program) earlier. The temporal variability in the execution of a similar program by MGE/PoA RGPs would further contribute to interneuron diversity (e.g., different chandelier cells or Martinotti cells).

Given that chandelier cells are among the last interneuron output by consecutively dividing MGE/PoA RGPs, their production would depend on the maintenance of RGPs following each round of division. Asymmetric neurogenic division of RGPs allows RGP self-renewal and, at the same time, neurogenesis directly or indirectly via intermediate progenitors that divide in the SVZ. Our in vivo analysis of individually dividing RGPs in the MGE/PoA showed that a large fraction of RGPs undergo asymmetric neurogenic division during interneuron neurogenesis. Interestingly, selective removal of PARD3, an evolutionarily conserved cell polarity protein previously implicated in neural progenitor cell asymmetric division[52,53], in MGE/PoA RGPs causes a switch from asymmetric neurogenic division to symmetric terminal division, and a loss of MGE/PoA RGPs at the late embryonic

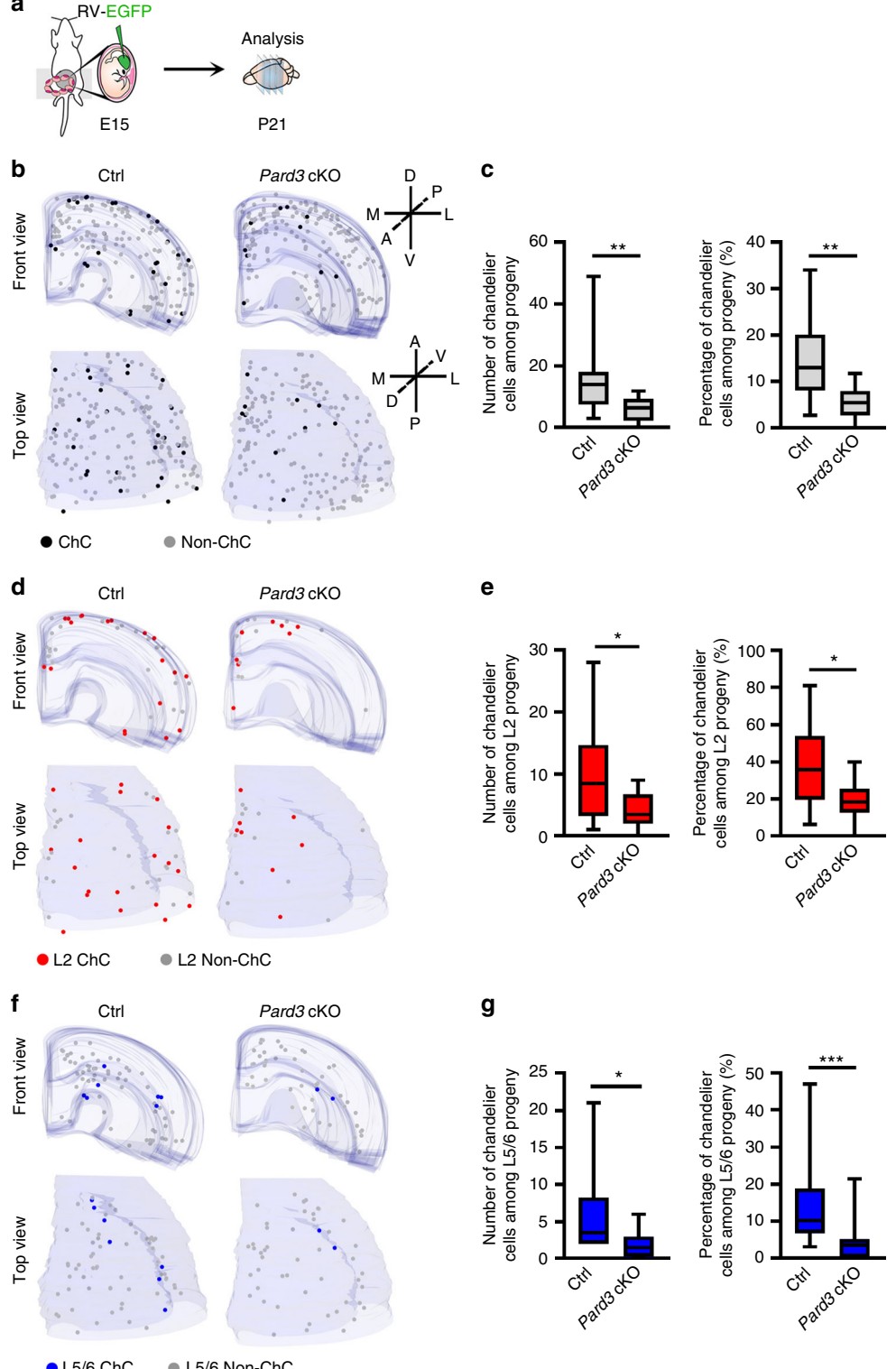

stage. Consequently, this leads to significant loss of both deep and superficial layer chandelier cells. These results suggest that proper RGP asymmetric division is essential for the generation of the correct number and types of neocortical interneurons. In other words, orderly progression of RGP asymmetric divisions supports the proper generation of different neocortical interneuron subtypes. The SVZ harboring IPs is greatly expanded in the MGE[54],

suggesting that IP division is also critical for the proper generation of neocortical interneurons. Consistent with this, a recent study suggested that both VZ and SVZ divisions affect neocortical interneuron production[63]. Notably, our analysis of interneuron output by dividing MGE/PoA RGPs at different embryonic stages includes the interneurons arising from their corresponding IP progeny as well.

**Fig. 7** Chandelier cell production depends on RGP asymmetric division. **a** Schematic of the experimental design. Animals received in utero injection of Cre-dependent EGFP-expressing retrovirus at E15 were analyzed at P21. **b** 3D reconstruction images of P21 control (Ctrl) and *Pard3* cKO neocortical hemispheres with EGFP-expressing chandelier cells and non-chandelier cells. Black and gray dots represent the cell bodies of EGFP-expressing chandelier cells and non-chandelier cells, respectively. A anterior, P posterior, D dorsal, V ventral, M medial, L lateral. **c** Quantification of the number (left) and percentage (right) of chandelier cells among EGFP-expressing interneurons per neocortical hemisphere (control, $n = 10$ hemispheres from 5 animals; *Pard3* cKO, $n = 16$ hemispheres from 8 animals). Center line, median; box, interquartile range; whiskers, minimum and maximum. ** $P = 0.003$ (Mann–Whitney test). **d** 3D reconstruction images of P21 control and *Pard3* cKO neocortical hemispheres with EGFP-expressing chandelier cells and non-chandelier cells in layer 2. Red and gray dots represent the cell bodies of EGFP-expressing chandelier cells and non-chandelier cells, respectively, in layer 2. **e** Quantification of the number (left) and percentage (right) of chandelier cells among EGFP-expressing interneurons in layer 2 per neocortical hemisphere (control, $n = 10$ hemispheres from 5 animals; *Pard3* cKO, $n = 16$ hemispheres from 8 animals). Center line, median; box, interquartile range; whiskers, minimum and maximum. *$P = 0.04$ (left); *$P = 0.03$ (right) (Mann–Whitney test). **f** 3D reconstruction images of P21 control and *Pard3* cKO neocortical hemispheres with EGFP-expressing chandelier cells and non-chandelier cells in layers 5/6. Blue and gray dots represent the cell bodies of EGFP-expressing chandelier cells and non-chandelier cells, respectively, in layers 5/6. **g** Quantification of the number (left) and percentage (right) of chandelier cells among EGFP-expressing interneurons in layers 5/6 per neocortical hemisphere (control, $n = 10$ hemispheres from 5 animals; *Pard3* cKO, $n = 16$ hemispheres from 8 animals). Center line, median; box, interquartile range; whiskers, minimum and maximum. * $P = 0.01$; *** $P = 0.0007$ (Mann–Whitney test)

A recent single-cell RNA-seq analysis of MGE RGPs failed to identify RGP subtypes based on gene expression[64]. This is consistent with the notion that a common, multipotent pool of MGE RGPs generate distinct interneuron subtypes. In line with this, previous clonal analyses showed that individual dividing MGE/PoA RGPs are capable of generating both PV- and SOM-expressing neocortical interneurons[36,65–67]. On the other hand, it has been suggested that the MGE can be subdivided into multiple domains based on the combinatorial expression of several transcriptions factors;[27,68] however, the relevance of these potential domains to neocortical interneuron diversity remains unclear. Previous fate-mapping and transplantation studies have suggested that the dorsal and ventral regions of the MGE exhibit a certain bias towards generating SOM-expressing and PV-expressing neocortical interneurons, respectively[23,68–71]. It is possible that some relatively fate-restricted RGPs may exist in the MGE/PoA[27], which cannot be effectively addressed by our current approach; nonetheless, our data suggest that diverse neocortical interneurons are progressively generated as a developmental lineage by consecutive divisions of multipotent RGPs. It lays the ground for future studies to better understand the developmental program that generates the appropriate number and diversity of inhibitory interneurons critical for the assembly of functional neural circuits in the neocortex.

## Methods

**Animals**. $R26^{LSL-TVAilacZ}$[37] and *Nkx2.1-Cre*[22] mouse lines were maintained at the Memorial Sloan-Kettering Cancer Center (MSKCC) animal facility. Timed pregnancies were set up between male $R26^{LSL-TVAilacZ}$ and female *Nkx2.1-Cre* mice. The plug date was designated as E0 and the date of birth was defined as P0. The *Pard3* knockout-first (*Pard3tm1a(KOMP)Wtsi*) ES cell line (EPD0334_1_C04) was obtained from the International Knockout Mouse Consortium (IKMC). After confirmation, ESC clones were injected into C57BL/6 J blastocysts, and the resulting chimeras were crossed with C57BL/6 J females to obtain germ-line transmission. The knockout-first allele was converted to the conditional allele by crossing with *B6.Cg-Tg(ACTFLPe)* mice (stock# 005703; The Jackson Laboratory) to excise the gene trap cassette. The resulting *Pard3fl/+* (fl, floxed allele) conditional mice were subsequently intercrossed to generate *Pard3fl/fl* mice. *Nkx2.1-Cre* transgenic mice were used to delete *Pard3* in the MGE/PoA. *Ai9-tdTomato* mice were obtained from The Jackson Laboratory (stock# 007909). All experimental procedures were approved by the Institutional Animal Care and Use Committee (IACUC) of the Memorial Sloan Kettering Cancer Center.

**Retrovirus production and in utero intraventricular injection**. RCAS retrovirus expressing EGFP or mCherry was produced as previously described[36]. The titer of the RCAS virus was ~10^{8-9} infectious units (IFU) per mL. Cre-dependent pUX-FLEX retrovirus expressing EGFP was produced as previously described[72]. In utero intraventricular retrovirus injection was performed as previously described[73]. In brief, the uterine horns of pregnant (E12-E17) *Nkx2.1-Cre;R26^{LSL-TVAilacZ}* mice were exposed. RCAS-EGFP retrovirus solution (~1.0 μL) with Fast Green (2.5 mg/mL, Sigma) was injected into the lateral ventricle through a beveled,

glass micropipette (Drummond Scientific). Following injection, the peritoneal cavity was lavaged with warm phosphate-buffered saline (PBS, pH 7.4), the uterine horns were replaced, and the wound was closed.

**Tissue preparation, immunohistochemistry, and imaging**. Embryonic or postnatal mice were transcardially perfused with ice-cold PBS (pH 7.4) followed by 4% paraformaldehyde (PFA) in PBS (pH 7.4). Following perfusion, the brains were dissected out and post-fixed in the same fixation solution overnight at 4 ºC. Serial coronal sections (70 μm) of each brain were prepared using a vibratome (Leica Microsystems). For immunohistochemistry, sections were incubated for 1 h at room temperature in a blocking solution (10% normal goat serum, 0.3% Triton X-100 in PBS), followed by incubation with the primary antibodies overnight at 4 ºC. The primary antibodies used were: rat monoclonal anti-GFP (Invitrogen, 1:1,000), chicken polyclonal anti-GFP (Aves, 1:1000); rabbit polyclonal anti-RFP (Rockland, 1:1,000), rat monoclonal anti-BrdU (Bio-Rad, 1:1,000), mouse monoclonal anti-Parvalbumin (Millipore, 1:500), rat monoclonal anti-Somatostatin (Millipore, 1:200), mouse antibody to Ki67 (BD Transduction Laboratories, 1:200), rabbit polyclonal anti-PARD3 (Sigma, 1:200), rabbit polyclonal anti-BLBP (Abcam 1:200), and rabbit polyclonal anti-OLIG2 (Millipore, 1:500), mouse monoclonal anti-OLIG2 (Millipore, 1:200). Sections were washed in 0.5% Triton X-100 in PBS and incubated with the appropriate secondary antibodies for 2 h at room temperature. Secondary antibodies used were: goat Aexa-488, Alexa-568, or Alexa 647 anti-rabbit, anti-rat, anti-chicken, or anti-mouse IgG (Invitrogen, 1:500). DNA was stained with 4',6-diamidino-2-phenylindole (DAPI, Invitrogen). For 3,3'-Diaminobenzidine (DAB, Invitrogen) staining, biotinylated goat anti-rabbit secondary antibody (Invitrogen, 1:500) was used. To enhance the signal, ABC solution (Vector Laboratories) was prepared 30 minutes prior to use and applied to sections for 1-2 h at room temperature before the DAB reaction. Sections were then mounted on 2% gelatin coated slides and coverslipped before imaging. Images were acquired with a laser scanning confocal microscope (FV1000, Olympus), and analyzed with FluoView (Olympus) and Photoshop (Adobe Systems).

**EdU and BrdU labeling**. For 5-ethynyl-2'-deoxyuridine (EdU) labeling, two injections of EdU (10 mg per kg body weight, Sigma) separated by 4 h were administered to timed pregnant mice via intraperitoneal injection. EdU was detected using Click-iT EdU imaging kit (Invitrogen). For 5-bromo-2'-deoxyuridine (BrdU) labeling, two injections of BrdU (50 mg per kg body weight, Sigma) 4 h apart were administered. BrdU was detected by first treating the sections with 1 N HCl for 1 h at 37 ºC and washing in 0.1 M borate buffer 3 times for 10 min, followed by immunostaining with rat monoclonal anti-BrdU as described above.

**Three-dimensional reconstruction and stereological analysis**. Serial coronal sections along the rostrocaudal axis were examined sequentially using Neurolucida (MBF Bioscience) installed on an upright microscope. The boundaries of the whole brain, cortex, hippocampus, striatum and midline were manually traced and aligned. Individual labeled interneurons were marked by colored dots. Based on the number of sections spanning the rostrocaudal axis of the brain, the total number of interneurons in each region (i.e., neocortex, hippocampus, and striatum), as well as subregion (i.e., neocortical layers), were calculated.

**In vivo RGP division pattern assay**. To sparsely label dividing RGPs at the VZ surface of the MGE/PoA, serially diluted, low-titer of replication-incompetent EGFP-expressing retrovirus was injected into the lateral ventricle of embryos in utero, as previously described[73,74]. At 24 h after the injection, embryonic brains were collected and serially sectioned using a cryostat (30 μm, Leica Microsystem). Consecutive sections covering the entire MGE/PoA were collected and stained with

antibodies against GFP, OLIG2, and Ki67. The entire MGE/PoA was reconstructed to recover all EGFP-labeled cell pairs by Neurolucida (MBF Bioscience) on an upright microscope equipped with epifluorescence illumination and cooled charged-coupled device camera (Zeiss).

**Statistical analysis**. Data were presented as median with interquartile range and whiskers as the minimum and maximum, or as mean ± sem. Statistical differences were determined using non-parametric Mann–Whitney test, Kruskal–Wallis test, Jonckheere-Terpstra test, unpaired $t$-test with Welch's correction, one-way ANOVA, or chi-square test. Statistical significance was set at $p < 0.05$. No statistical method was used to predetermine sample size. At least 3 animals per genotype have been analyzed for each experiment. No data points were excluded. All Samples per genotype were randomly allocated into experimental groups. The investigators were not blinded to group allocation. The variances between the groups were not being statcially caompared.

## Data availability

The datasets generated during the current study are available from the corresponding authors on reasonable request.

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

## Acknowledgements

We thank Dr. Shaoyu Ge (Stony Brook University) for retroviral plasmid, and Dr. Margaret E. Ross (Weill Cornell Medical College) and the members of the Shi laboratory for valuable discussion and input. This work was supported by grants from the NIH (R01DA024681, R01MH101382, R01NS105477, and R01NS085004 to S.-H.S., and P30CA008748 to Memorial Sloan Kettering Cancer Center Core Facilities), the Human Frontier Science Program (RGP0053/2014 to S.-H.S.), the New York State Stem Cell Science (NYSTEM) grant (N13G-232 to S.-H.S.), the Starr Foundation Tri-Institutional Stem Cell Initiative Grant (to S.-H.S.), and the Howard Hughes Medical Institute (to S.-H.S.).

## Author contributions

K.T.S. and S.-H.S. conceived the project; K.T.S. performed the stereological analysis of interneuron output with help from X.-J.Z. and O.D.; W.A.L. generated and analyzed *Pard3* mutant mice; Z.-L.L., Z.S., and J.M. performed morphological and immunohistochemical characterization; Z.L. provided retroviruses and helped with mouse colony management; K.T.S. and S.-H.S. wrote the paper with input from all other authors.

## Additional information

**Competing interests:** The authors declare no competing interests.

