## [Peer Review File · Nature Communications]

Reviewers' comments:

Reviewer #1 (Remarks to the Author):

I think the authors have improved the manuscript and have addressed my experimental concerns in a convincing manner.

However, they over emphasize their findings throughout the manuscript. One blatant example, pointed out by both me and Reviewer 2 is that it is pretty clear that the current findings largely confirm those in Miyoshi et al 2007 using slightly different methods. Instead of acknowledging this the authors choose to maliciously re-interpret the findings from the Olig2-CreER:Z/EG claiming that the temporal specificity reported by Miyoshi was not true. In my mind, the current study shows that, in addition to the temporal code shown previously, this is not from separate RGPs but from a single lineage as also suggested by reference 67.

Another example of many is on line 88: "This is especially pertinent for neocortical interneurons, as the developmental origin of their diversity at the progenitor level has not been systematically investigated." - In the discussion the authors then cite several papers doing exactly this to pretty much the same extent as the current study.

I like this study, it is elegant, timely and convincing. I am sorry to say that the aggressive tone of the writing significantly reduces the overall impression and unless the authors temper their claims I cannot support its publication in Nature Communications.

Reviewer #2 (Remarks to the Author):

The manuscript by Sultan and colleagues describes a number of histological fate-mapping experiments to explore the progeny of ganglionic eminence progenitors in the mammalian telencephalon.

I have reservations about the extent of the conceptual advance made by this manuscript. The authors propose that the key findings are: (1) that MGE progenitor pool produces distinct interneuron progeny over development that populate the cortical plate in an inside out manner with a late outside in distribution; (2) Chandelier cells are born at late stages from progenitor that produce other subtypes earlier in development; (3) PARD3 leads to depletion of the progenitor pool and loss of ChCs.

The first point is well established in the field of interneuron development since the discovery of distinct neurogenic niches in the ventral ganglionic eminences. A number of groups have shown that different interneuron subtypes are generated over time and populate the cortex

in a progressive inside-out manner notably Miyoshi et al. (2007) and Rymar & Sadikot (2007). This point is well accepted by the community and covered in numerous reviews, for example Bartolini et al. (2013). The later outside-in pattern is not at all surprising given the existing evidence that Chandelier cells are born late from Nkx2-1 progenitors (Taniguchi et al., 2013). Evidence presented in that paper (Figure 2) clearly identifies that Chandelier cells are preferentially located in layers 2 and 6 (identical to the data shown in Figure 1a and 1d of the present manuscript). Finally the current group have already published on the role of PARD3 in regulating asymmetric division in neocortex (Bultje et al., 2009), a paper which also shows PARD3 protein expression in the ganglionic eminences. While of course it is important to bring all of these strands together in a cohesive argument – in a well presented and detailed manner – it is hard to see how exactly this furthers our knowledge in a way that would be appreciated by the broad readership of a journal such as Nature Communications.

The characterization of interneuron diversity is lacking throughout the manuscript. The authors test for both PV and SOM (page 7) but only make a limited attempt to quantify the numbers at a later stage (p.10, Supplementary Figure 5). On pages 9-10 (lines 198+) the authors describe the morphology of various interneuron subtypes but terms such as 'dense arbor cell' and 'horizontal cell' are not widely recognized in the GABAergic interneuron community. It would be far more useful if immunohistochemistry had been combined with morphology to provide a more detailed and meaningful appraisal of interneuron diversity. The authors are correct in that the one of the few cell types that can be meaningfully recognized based on morphology alone are Chandelier cells.

On a number of occasions the authors are over-enthusiastic in the language used. Two examples: on line 50 the authors claim that the diverse inhibitory system has 'extraordinary' power. It would be more reasonable to replace 'extraordinary' with 'requisite'. Again on lines 96-97 it would be more appropriate to suggest that Chandelier cells control pyramidal cells through the release of GABA rather than stipulating 'powerful inhibitory control'.

Reviewer #3 (Remarks to the Author):

Although the diversity and functional significance of cortical GABAergic interneurons are well recognized, the developmental mechanisms and logic of their production remain poorly understood.

Previous studies have identified the embryonic regions (e.g. MGE, POA) and key transcription factors (e.g. Nkx2.1, Lhx6) involved in the generation of major cohorts of interneurons. However, these studies are at a coarse resolution both in terms of progenitors and interneurons – they do not recognize the different types of progenitors, their lineage progress, and their relationship to distinct interneuron cell types. To date, a coherent framework of cortical interneuron generation and specification with cellular and lineage resolution is yet to emerge. In this context, the work by Sultun et al has the potential to

make a major contribution. They used a retroviral method that primarily targets MGE radial glia progenitors (RGPs). They use cell birth dating and double time infection to track the lineage progression of these RGPs. They use whole brain serial sectioning to quantify the laminar pattern of interneurons generated from E12-E17. And they focus on chandelier cells (ChCs), the most well recognizable interneuron cell type, which allowed them to gain important insight on the patterns of MGE neurogenesis. Together, they provide evidence that MGE RGPs divide progressively to generate multiple cohorts of interneurons which occupy the cortex in an inside-out-inside pattern. In addition, ChCs are generated near the end of RGP lineage progression that first generates non-ChCs, and interestingly layer 2 ChCs are generated in similar proportions during their production period, suggestive of a predictable neurogenic program. However, there are a number of areas, both in terms of experiments and presentation that need to be addressed to fulfill the potential of this work.

1. Among the main figures, Figure 2 is rather descriptive and weak, and can be moved as a supplement figure. In a, the naming of "dense arbor", "descending axon", and "horizontal cells" are necessary but unconventional, again better suited in supplement. The identification of Martinotti cells is not entirely convincing. In E14-E21 column, L4 "Martinottie cell", the vertical neurite seems too thick to be an axon, and has no or few layer branches characteristic to Martinottie cells.

2. The authors strived for a comprehensive fate mapping of Nkx2.1 RGPs from E12 to E17. However, Nkx2.1 expression and MGE formation begin by E10 with onset of neurogenesis. Importantly, there may be differences in the properties of early and late cohorts of MGE progenitor types including RGPs. It may be difficult to use the current method to probe these early progenitors. The authors can discuss this point more thoroughly in page 22 after line 482.

3. The current supplement Figure 6 addresses an important issue and can be moved to main figure if it can be enhanced.

4, the use of "temporally progressive" throughout the text is redundant – progressive is mostly temporal.

5. the use of the word "progressive *evolution* of a common multipotent RGP" is not appropriate - *evolution" should be replaced.

5. In page 11 first paragraph, line 232-237 statements are misleading. fate-restricted progenitors can generate the same cell type at successive divisions, thus these same cell type progeny are born at different times. Conversely at least in invertebrates, e.g. drosophila, the same progenitor through asymmetric division can generate two cell types born at the same time. So these statements should be modified or removed. They are not really needed to introduce this section.

6. page 13, line 272 ", even though chandelier cells have been recently suggested to be selectively generated by NKX2.1 progenitor cells ... at the late embryonic stage". This second half sentence appears to contradict the first half, when it is not (as stated clearly in

the next paragraph). Put this half sentence here is misleading. This can be moved to the next paragraph.

7. page 15 line 321-325. This statement is misleading and appears contradictory to the previous result that E15 is the peak chandelier cell production time (page 13, line 282). What the author meant here, I think, is that the CUMULATIVE production of chandelier cells from a particular labeling time (E12, E13, ...). This should be made very clear.

8. page 21 line 463, change to "... undergo several rounds of divisions before producing superficial layer chandelier cells at the later time point".

9. page 24 line 531-534. The issue of potentially fate-restricted MGE progenitors in different spatial domains is an important one that cannot be address with the current viral labeling method. I suggest to add "and our current method cannot address this issue" following " ... may exist in the MGE/PoA".

Point-by-point Responses to Reviewers' Comments (NCOMMS-18-18261-T)

Reviewer #1 (Remarks to the Author):

I think the authors have improved the manuscript and have addressed my experimental concerns in a convincing manner.

*However, they over emphasize their findings throughout the manuscript. One blatant example, pointed out by both me and Reviewer 2 is that it is pretty clear that the current findings largely confirm those in Miyoshi et al 2007 using slightly different methods. Instead of acknowledging this the authors choose to maliciously re-interpret the findings from the *Olig2-CreER;Z/EG* claiming that the temporal specificity reported by Miyoshi was not true. In my mind, the current study shows that, in addition to the temporal code shown previously, this is not from separate RGP's but from a single lineage as also suggested by reference 67.*

Response: We thank the reviewer for considering that our revision has “*improved the manuscript and addressed the experimental concerns in a convincing manner*”.

By no means did we have any intention to claim that the temporal specificity reported by Miyoshi was not true. In the previous round of review, the reviewer specifically stated that “...*but if I remember correctly this story (Miyoshi et al) as actually the opposite in that the *Olig2-CreER;Z/EG* strategy did NOT label the RGP's – but only the progeny of the last division. Thus, that data actually in some ways argues against the authors claims*”. To address this, we explicitly examined the identity of the cells labeled by *Olig2-CreER* by performing both immunohistochemistry and genetic tracing experiments (**Supplementary Fig. 8**). Our data clearly suggest that *Olig2-CreER* labels BLBP⁺ RGP's in the VZ of the MGE as well as the LGE and the CGE (consistent with the labeling of VIP⁺ interneurons in the neocortex reported in Miyoshi et al.), but NOT only the progeny of the last division.

It is important to note that Miyoshi et al. did not examine the division time of labeled progenitors (i.e., RGP's) or the birth date of labeled interneurons. In comparison, our study directly examined the division time of labeled NKX2.1⁺ MGE/PoA RGP's (**Fig. 1**) and the birth date of labeled interneurons (**Fig. 2**). Our data suggest an initial inside-out interneuron generation by the MGE/PoA NKX2.1⁺ RGP's, consistent with the overall interpretation by Miyoshi et al., as we stated in the text (**page 11-12**). Related to this, Miyoshi et al. explicitly stated that the late-labeled populations with tamoxifen administration at E15.5 enriched within the superficial layers of the cortex arise “from progenitors that do not express NKX2.1”. Our data showed for the first time that NKX2.1⁺ RGP's continue to produce neocortical interneurons in an outside-in manner after E15.5.

We revised the related sentence to “..., consistent with the previous genetic fate

mapping analysis of RGPs in the ventral eminence using the *Olig2-CreER* transgenic mice⁴¹ (**Supplementary Fig. 8a**).” (page 11-12)

Another example of many is on line 88: “This is especially pertinent for neocortical interneurons, as the developmental origin of their diversity at the progenitor level has not been systematically investigated.” - In the discussion the authors then cite several papers doing exactly this to pretty much the same extent as the current study.

Response: We revised this sentence to “This is especially pertinent for neocortical interneurons, as the developmental mechanisms and logic of their production at the progenitor level are not well understood.” (page 5)

I like this study, it is elegant, timely and convincing. I am sorry to say that the aggressive tone of the writing significantly reduces the overall impression and unless the authors temper their claims I cannot support its publication in Nature Communications.

Response: We toned down our findings in the revision.

Reviewer #2 (Remarks to the Author):

The manuscript by Sultan and colleagues describes a number of histological fate-mapping experiments to explore the progeny of ganglionic eminence progenitors in the mammalian telencephalon.

Response: We thank the reviewer for valuable and constructive comments.

I have reservations about the extent of the conceptual advance made by this manuscript. The authors propose that the key findings are: (1) that MGE progenitor pool produces distinct interneuron progeny over development that populate the cortical plate in an inside out manner with a late outside in distribution; (2) Chandelier cells are born at late stages from progenitor that produce other subtypes earlier in development; (3) PARD3 leads to depletion of the progenitor pool and loss of ChCs.

The first point is well established in the field of interneuron development since the discovery of distinct neurogenic niches in the ventral ganglionic eminences. A number of groups have shown that different interneuron subtypes are generated over time and populate the cortex in a progressive inside-out manner notably Miyoshi et al. (2007) and Rymar & Sadikot (2007). This point is well accepted by the community and covered in numerous reviews, for example Bartolini et al. (2013). The later outside-in pattern is not at all surprising given the existing evidence that Chandelier cells are

born late from Nkx2-1 progenitors (Taniguchi et al., 2013). Evidence presented in that paper (Figure 2) clearly identifies that Chandelier cells are preferentially located in layers 2 and 6 (identical to the data shown in Figure 1a and 1d of the present manuscript). Finally the current group have already published on the role of PARD3 in regulating asymmetric division in neocortex (Bultje et al., 2009), a paper which also shows PARD3 protein expression in the ganglionic eminences. While of course it is important to bring all of these strands together in a cohesive argument – in a well presented and detailed manner – it is hard to see how exactly this furthers our knowledge in a way that would be appreciated by the broad readership of a journal such as Nature Communications.

Response: In our study, we, for the first time, selectively and systematically examined the neocortical interneuron output by DIVIDING NKX2.1⁺ RGPs in the MGE/PoA at different embryonic stages (E12-E17). In comparison, the previous studies including Miyoshi et al and Rymar & Sadikot did not specifically link the exact timing of RGP division to diverse neocortical interneuron output, as the labeling approaches by Miyoshi et al and Rymar & Sadikot could not distinguish dividing vs. non-dividing progenitor cells or the types of dividing progenitors (e.g., RGPs vs. intermediate progenitors, IPs). In addition, *Olig2-CreER* labels a majority of the ventral telencephalon, including the MGE/PoA, the LGE, and the CGE (the primary origin of VIP⁺ interneurons; consistent with the labeling of VIP⁺ interneurons in Miyoshi et al.).

While the inside-out interneuron generation has been inferred previously (e.g., Miyoshi et al.), it remains largely unclear whether interneurons in deep layers are produced by RGPs that DIVIDE at the early embryonic stage, whereas interneurons in superficial layers are generated by RGPs that DIVIDE at the late embryonic stage.

Furthermore, to the best of our knowledge, the late outside-in interneuron generation by NKX2.1⁺ MGE/PoA RGPs has never been previously suggested. While chandelier cells have been shown to be born at the late embryonic stage (Taniguchi et al., 2013; Inan et al., 2012), two important questions remain unanswered: 1) Are chandelier cells generated by a dedicated/specific progenitor pool that divide preferentially at the late embryonic stage or by a common progenitor pool that divide progressively to produce non-chandelier cells at the early embryonic stage and then to produce chandelier cells at the late embryonic stage? 2) Are chandelier cells in different layers generated in an inside-out or outside-in manner? In other words, it is unclear whether deep layer chandelier cells are generated before or after superficial layer chandelier cells. Our study provides important new insights into these two fundamental questions.

While PARD3 has been shown to regulate RGP asymmetric division in the developing cortex, its role in regulating RGP division in MGE/PoA and interneuron generation has not been examined. More importantly, the emphasis of PARD3 in this study is not simply on its function in regulating MGE/PoA RGP division, but to use it as a

molecular tool to perturb RGP asymmetric division so as to directly test the hypothesis that consecutive/continuous MGE/PoA RGP asymmetric division is essential for RGP maintenance and late production of chandelier cells.

In summary, our findings directly support a progressive interneuron fate specific program via consecutive asymmetric division of multipotent MGE/PoA RGPs to generate diverse neocortical interneurons, among which chandelier cells are the last output. This program has not been previously suggested and will serve as a basic framework for understanding diverse neocortical interneuron generation.

The characterization of interneuron diversity is lacking throughout the manuscript. The authors test for both PV and SOM (page 7) but only make a limited attempt to quantify the numbers at a later stage (p.10, Supplementary Figure 5). On pages 9-10 (lines 198+) the authors describe the morphology of various interneuron subtypes but terms such as 'dense arbor cell' and 'horizontal cell' are not widely recognized in the GABAergic interneuron community. It would be far more useful if immunohistochemistry had been combined with morphology to provide a more detailed and meaningful appraisal of interneuron diversity. The authors are correct in that the one of the few cell types that can be meaningfully recognized based on morphology alone are Chandelier cells.

Response: In this study, we focused on chandelier cells, a well-defined interneuron subtype, to demonstrate the temporal program of diverse interneuron generation by multipotent MGE/PoA RGPs. It will be interesting in the future to systematically characterize different interneuron subtypes labeled in our study using a combination of approaches, including morphology, immunohistochemistry, and electrophysiology. However, in our view, this is beyond the scope of this study.

To further address the reviewer's point, we carried out new experiments to link our morphological characterization of the major representative interneurons to PV and SOM expression (**Supplementary Fig. 7**). We found that Martinotti cells and dense arbor cells are predominantly SOM⁺, whereas basket cells and chandelier cells are mostly PV⁺, consistent with the previous observations (Markram 2004; Ma et al., 2006; Tremblay R et al., 2016; Urban-Ciecko et al., 2016; Kubota et al., 2016; Paul, et al., 2017).

We moved the data of morphological characterization to Supplementary Information (**Supplementary Fig. 5**).

On a number of occasions the authors are over-enthusiastic in the language used. Two examples: on line 50 the authors claim that the diverse inhibitory system has 'extraordinary' power. It would be more reasonable to replace 'extraordinary' with 'requisite'. Again on lines 96-97 it would be more appropriate to suggest that Chandelier cells control pyramidal cells through the release of GABA rather than

stipulating ‘powerful inhibitory control’.

Response: Following the reviewer's suggestions, we replaced “extraordinary” with “requisite” (page 3), and “powerful inhibitory control” with “control pyramidal cell activity through the release of GABA” (page 5).

Reviewer #3 (Remarks to the Author):

Although the diversity and functional significance of cortical GABAergic interneurons are well recognized, the developmental mechanisms and logic of their production remain poorly understood.

Previous studies have identified the embryonic regions (e.g. MGE, POA) and key transcription factors (e.g. Nkx2.1, Lhx6) involved in the generation of major cohorts of interneurons. However, these studies are at a coarse resolution both in terms of progenitors and interneurons – they do not recognize the different types of progenitors, their lineage progress, and their relationship to distinct interneuron cell types. To date, a coherent framework of cortical interneuron generation and specification with cellular and lineage resolution is yet to emerge. In this context, the work by Sulton et al has the potential to make a major contribution. They used a retroviral method that primarily targets MGE radial glia progenitors (RGPs). They use cell birth dating and double time infection to track the lineage progression of these RGPs. They use whole brain serial sectioning to quantify the laminar pattern of interneurons generated from E12-E17. And they focus on chandelier cells (ChCs), the most well recognizable interneuron cell type, which allowed them to gain important insight on the patterns of MGE neurogenesis. Together, they provide evidence that MGE RGPs divide progressively to generate multiple cohorts of interneurons which occupy the cortex in an inside-out-inside pattern. In addition, ChCs are generated near the end of RGP lineage progression that first generates non-ChCs, and interestingly layer 2 ChCs are generated in similar proportions during their production period, suggestive of a predictable neurogenic program. However, there are a number of areas, both in terms of experiments and presentation that need to be addressed to fulfill the potential of this work.

Response: We thank the reviewer for considering our study “has the potential to make a major contribution” and for providing valuable and constructive comments.

1. Among the main figures, Figure 2 is rather descriptive and weak, and can be moved as a supplement figure. In a, the naming of “dense arbor”, “descending axon”, and “horizontal cells” are necessary but unconventional, again better suited in supplement. The identification of Martinotti cells is not entirely convincing. In E14-E21 column, L4 “Martinottie cell”, the vertical neurite seems too thick to be an

axon, and has no or few layer branches characteristic to Martinotti cells.

Response: Following the reviewer's suggestion, we moved the original Fig. 2 to **Supplementary Fig. 5**.

We identified Martinotti cells based on the previously established criteria of possessing prominent upward projecting axons, especially in layer 1. To address the reviewer's concern, we replaced E14-P21 L4 Martinotti cell with a new example with extensive thin axonal branches in superficial layers (**Supplementary Fig. 5, middle**).

2. The authors strived for a comprehensive fate mapping of Nkx2.1 RGP from E12 to E17. However, Nkx2.1 expression and MGE formation begin by E10 with onset of neurogenesis. Importantly, there may be differences in the properties of early and late cohorts of MGE progenitor types including RGP.

It may be difficult to use the current method to probe these early progenitors. The authors can discuss this point more thoroughly in page 22 after line 482.

Response: We thank the reviewer for this important point. In our study, we focused on E12-17, the peak phase of neocortical interneuron generation by MGE/PoA RGP. It is indeed technically difficult to perform our analysis at E10.

Our data suggest that MGE/PoA RGP progress through a temporal neurogenic program in generating diverse neocortical interneurons at different embryonic stages. Dividing MGE/PoA RGP prior to E12 likely exhibit a similar temporal program in producing non-chandelier cells initially and chandelier cells later. It is possible that the MGE/PoA RGP starting the temporal neurogenic program earlier (e.g., undergo neurogenic division at E10 to produce interneurons) also complete the program (i.e., exit the cell cycle or neurogenic program) earlier. The temporal variability in the execution of a similar neurogenic program by MGE/PoA RGP would further contribute to interneuron diversity (e.g., different chandelier cells or Martinotti cells).

We included this in-depth discussion (**page 22**).

3. The current supplement Figure 6 addresses an important issue and can be moved to main figure if it can be enhanced.

Response: Following the reviewer's suggestion, we moved the original Supplementary Fig. 6 to **Fig. 2**.

4, the use of "temporally progressive" throughout the text is redundant – progressive is mostly temporal.

Response: We revised accordingly by removing "temporally" throughout the text.

5. the use of the word “*progressive *evolution* of a common multipotent RGP*” is not appropriate - **evolution**” should be replaced.

Response: We revised “evolution” to “change” (page 10).

6. In page 11 first paragraph, line 232-237 statements are misleading. *fate-restricted progenitors can generate the same cell type at successive divisions, thus these same cell type progeny are born at different times. Conversely at least in invertebrates, e.g. drosophila, the same progenitor through asymmetric division can generate two cell types born at the same time. So these statements should be modified or removed. They are not really needed to introduce this section.*

Response: We removed the sentence (page 11).

7. page 13, line 272 “, even though chandelier cells have been recently suggested to be selectively generated by NKX2.1 progenitor cells ... at the late embryonic stage”. This second half sentence appears to contradict the first half, when it is not (as stated clearly in the next paragraph). Put this half sentence here is misleading. This can be moved to the next paragraph.

Response: We removed the sentence (page 12).

8. page 15 line 321-325. This statement is misleading and appears contradictory to the previous result that E15 is the peak chandelier cell production time (page 13, line 282). What the author meant here, I think, is that the CUMULATIVE production of chandelier cells from a particular labeling time (E12, E13, ...). This should be made very clear.

Response: We added “cumulative” to the sentence (page 15).

9. page 21 line 463, change to “... undergo several rounds of divisions before producing superficial layer chandelier cells at the later time point”.

Response: We revised accordingly (page 21).

10. page 24 line 531-534. The issue of potentially fate-restricted MGE progenitors in different spatial domains is an important one that cannot be address with the current viral labeling method. I suggest to add “and our current method cannot address this issue” following “ ... may exist in the MGE/PoA”.

Response: We revised accordingly (page 24).